

# Chemically Speciated Air Pollutant Emissions from Open Burning of Household Solid Waste from South Africa

Xiaoliang Wang[1], Hatef Firouzkouhi[1], Judith C. Chow[1], John G. Watson[1], Steven Sai Hang Ho[1], Warren Carter[2], Alexandra S.M. De Vos[2]

[1] Division of Atmospheric Sciences, Desert Research Institute, Reno, NV 89512, U.S.A.
[2] SASOL Research and Technology, Sasolburg, South Africa

*Correspondence to*: Xiaoliang Wang (xiaoliang.wang@dri.edu)

## Abstract

Open burning of household solid waste is a large source of air pollutants worldwide, especially in developing countries. However, waste burning emissions are either missing or have large uncertainties in local, regional, or global emission inventories due to limited emission factor (EF) and activity data. Detailed particulate matter (PM) chemical speciation data is even scarcer. This paper reports source profiles and EFs for $PM_{2.5}$ species as well as acidic and alkali gases measured from laboratory combustion of ten waste categories that represent open burning in South Africa. Carbonaceous materials contributed

more than 70% of $PM_{2.5}$ mass. Elemental carbon (EC) was most abundant from flaming materials (e.g., plastic bags, textile, and combined materials) and its climate forcing exceeded the corresponding $CO_2$ emissions by a factor of 2–5. Chlorine had the highest EFs among elements measured by X-ray fluorescence (XRF) for all materials; vegetation emissions showed high abundances of potassium, consistent with its use as a marker for biomass burning. Fresh $PM_{2.5}$ emitted from waste burning appeared to be acidic. Moist vegetation and food discards had the highest hydrogen fluoride (HF) and PM fluoride EFs due to

fluorine accumulation in plants, while burning rubber had the highest hydrogen chloride (HCl) and PM chloride EFs due to high chlorine content in the rubber. Plastic bottles and bags, rubber, and food discards had the highest EFs for polycyclic aromatic hydrocarbons (PAHs) and nitro-PAHs as well as their associated toxicities. Distinct differences between odd and even carbon preferences were found for alkanes from biological and petroleum-based materials: dry vegetation, paper, textile, and food discards show preference for the odd-numbered alkanes, while the opposite is true for plastic bottles, bags, and rubber.

As phthalates are used as plasticizers, their highest EFs were found for plastic bottles and bags, rubber, and combined materials. Data from this study will be useful for health and climate impact assessments, speciated emission inventories, source-oriented dispersion models, and receptor-based source apportionment.





## 1 Introduction

Uncontrolled open burning is a common practice to dispose of household or municipal solid waste (MSW) in many rural
communities, especially in developing countries (Cook and Velis, 2021; Sharma et al., 2022; Okedere et al., 2019; Das et al.,
2018; Bulto, 2020; Reyna-Bensusan et al., 2018; Cheng et al., 2020). Open burning has low combustion efficiencies due to
inefficient mixing of fuels and oxygen and low burning temperatures, resulting in emissions of a wide range of air pollutants.
MSW is often burned close to community residences. The limited dispersion and dilution increase direct inhalation exposures
and exacerbate adverse health effects (Wiedinmyer et al., 2014; Lemieux et al., 2004; Krecl et al., 2021). MSW open burning
emissions deteriorate air quality in neighborhood-, urban-, and regional-scales (Oleniacz et al., 2023). Communities with lower
socioeconomic status are often more impacted by MSW burning emissions, leading to environmental justice concerns
(Nagpure et al., 2015; Martuzzi et al., 2010). It is estimated that exposure to $PM_{2.5}$ from open burning of solid waste causes at
least 270,000 premature deaths per year globally (Williams et al., 2019). Open burning also contributes to climate change as a
result of large carbon dioxide ($CO_2$) and light absorbing carbon (including black carbon [BC]) emissions, two of the largest
climate forcers (IPCC, 2013; Reyna-Bensusan et al., 2018; Bond et al., 2013).

Despite the large environmental impacts of uncontrolled MSW open burning, its emissions are not included or are poorly
represented in local, regional, and global emission inventories due to lack of emission factor (EF) and activity data (Cook and
Velis, 2021; Ramadan et al., 2022). Most existing inventories only include criteria pollutants (U.S. EPA, 2023) or greenhouse
gases (IPCC, 2006) with limited chemical speciation. In addition to criteria pollutants, solid waste burning emits other toxic
compounds. Construction timber combustion releases high concentrations of arsenic (As), chromium (Cr), and copper (Cu),
while plastic burning releases phthalates, polycyclic aromatic hydrocarbons (PAHs), and dioxins (Velis and Cook, 2021;
Wasson et al., 2005). Lemieux (1997, 1998) reported gas and speciated particle emissions from simulated open burning of
household waste in barrels. However, the study was limited to testing U.S. households with and without recycling practices.
Barrel burning may not represent open pile burning due to different fuel-air interactions. Lemieux et al. (2004) further
summarized organic air toxics from open burning of many materials, including MSW, and these data are used in the US
Environmental Protection Agency (US EPA) AP-42 Compilation of Air Emissions Factors (U.S. EPA, 1992). Stockwell (2016)
measured emissions from laboratory burning of shredded tires, plastic bags, mixed waste, and a variety of biomass species.
Gaseous chemical EFs were presented, but PM chemistry was not reported. Jayarathne et al. (2018) reported EFs for $PM_{2.5}$
(particles with aerodynamic diameter ≤2.5 µm) and several components (carbon, ions, metals, and organics) for combustion
sources in Nepal, including mixed waste under dry and damp conditions, two types of plastic mixtures, and crop residues.
Emissions of criterial pollutants and $PM_{2.5}$ compositions (carbon, ions, and metals) were reported for several types of MSW in
China (Cheng et al., 2020). Several studies characterized PM mass and chemistry for plastics burning (Hoffer et al., 2020; Wu
et al., 2021). These studies highlighted the large variation of EFs due to the heterogeneities in waste compositions and burning
conditions.

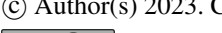



Detailed PM chemical compositions are important for many applications. The association between PM mass exposure and mortality is well established and exposure to $PM_{2.5}$ is one of the most important risk factors for premature death in the global burden of disease (Murray et al., 2020). However, as PM is a complex mixture, the toxicity and carcinogenicity of different chemical species are still uncertain (Lighty et al., 2000; Kelly and Fussell, 2012). Epidemiology and toxicology studies of PM chemicals are needed to develop a mechanistic understanding of their health effects. PM compositions are also used to evaluate

visibility and climate effects. PM light scattering and absorption properties depend on its chemical composition and associated hygroscopicity and optical properties (Watson, 2002). BC, or elemental carbon (EC), is the major light absorbing component of PM and may have a global warming potential 900 times that of $CO_2$ (Bond et al., 2013). Speciated emission inventories have been applied in source-oriented dispersion models to estimate ambient concentrations and deposition patterns and to target effective emission reduction strategies (Reff et al., 2009; Simon et al., 2010). Deposition from MSW burning emissions

is a major cause of discoloring of the Taj Mahal in India (Lal et al., 2016). PM chemical composition is necessary for receptor-based source apportionment, which uses the chemical abundance patterns in source profiles to quantify contributions of different sources to ambient PM concentrations (Watson et al., 2016). The source profile collinearities caused by similarities is a major cause of source apportionment uncertainty. Extending chemical analyses beyond conventionally analyzed elements and ions to include particle-phase organics can potentially provide molecular markers to minimize collinearities.

This study used comprehensive laboratory combustion experiments to quantify emissions from ten types of MSW from South Africa. EFs for $CO_2$ and criteria air pollutants have been reported by Wang et al. (2023). This paper focuses on speciated source profiles and EFs including elements, acidic and alkali gases and ions, PAHs, nitro-PAHs, n-alkanes, and phthalates.

## 2 Methodology

### 2.1 Waste Materials and Combustion Experiments

MSW materials were collected from typical household refuses by SASOL, a petrochemical and energy company, in the Zamdela community in South Africa. This is part of SASOL's Waste Collection Interventions (WCI) program to assist local communities in MSW collection and disposal in landfills to reduce illegal open burning and improve air quality. The materials were sent to the Desert Research Institute (DRI) in Reno, Nevada, USA for emission testing. Food discards and vegetation samples were collected in Nevada to avoid deterioration during shipping. The ten types of waste categories tested include: 1)

paper; 2) leather/rubber; 3) textile; 4) hard plastic bottles and food containers; 5) soft plastic bags; 6) dry vegetation (0% moisture content); 7) natural vegetation (20% moisture content); 8) damp vegetation (50% moisture content); 9) food discards; and 10) combined materials. The combined materials were mixtures of all categories based on their mass fractions in MSW. Glass, metals, and ceramics were added to the combination to mimic their influences on burning emissions. Moisture contents were measured right after collection in the field. Before testing, the waste materials (except food discards) were oven dried at

90 °C for 24 hours, rehydrated to their natural moisture levels with distilled deionized water (DDW), and re-equilibrated for at least 24 hours.



The major elemental compositions (i.e., carbon [C], hydrogen [H], nitrogen [N], sulfur [S], and oxygen [O]) of the waste materials were measured by an elemental analyzer (Model Flash EA1112, Thermo Scientific). The carbon content was used for the fuel-based EF calculation. The same procedure was used to quantify the elemental compositions of ashes after combustion.

The experimental method has been documented by Wang et al. (2023) and only a brief description is provided here. For each burn, 0.5 to 20 g of waste material was placed in a ceramic crucible and maintained at 450 °C to simulate open burning. Each burn typically took 1800 s, varying from 1000 to 4000 s. Paper, textile, soft plastic bags, vegetations (with dry and natural moisture contents), and combined waste had both flaming and smoldering phases, while leather/rubber, plastic bottles, damp vegetation, and food discards only smoldered. A suite of gas and particle analyzers monitored the concentrations in real time, including $CO_2$, carbon monoxide (CO), oxides of nitrogen ($NO_x$), sulfur dioxide ($SO_2$), $PM_{2.5}$, and $PM_{10}$ (particles with aerodynamic diameter ≤10 μm). Integrated PM samples were collected simultaneously onto four parallel filters that accommodated different chemical analyses (Figure 1). A total of 29 filter sets were collected, including three replicates for each of the eight fuels, two replicates for vegetations with 20% and 50% moisture content, and one field blank.

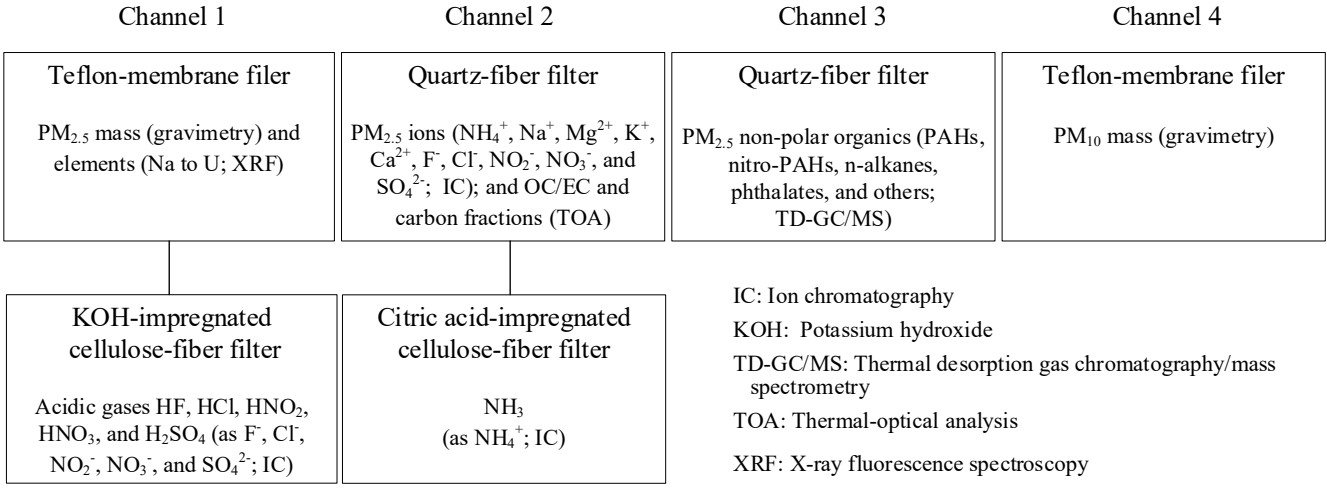

Figure 1: Analyses for the three $PM_{2.5}$ and one $PM_{10}$ filter channels.

## 2.2 Chemical Analysis

As shown in Figure 1, the $PM_{10}$ filters (Channel 4) were weighed for gravimetric mass concentrations, while the three $PM_{2.5}$ filters (Channels 1–3) were analyzed for elements, ions, carbon fractions, and organic compounds (Chow and Watson, 2013). Back-up filters impregnated with gas-absorbing solutions were analyzed for acidic and alkali gases.

Channel 1 is a Teflon-membrane filter backed by a potassium hydroxide (KOH)-impregnated cellelose-fiber filter. The Teflon-membrane filters were analyzed for $PM_{2.5}$ mass by gravimetry using microbalances (Model XP-6, Mettler Toledo, Hightstown, NJ) with 1 μg sensitivity (Watson et al., 2017). In addition, 51 elements (from Na to U) were measured using a



PANalytical X-ray fluorescence (XRF) analyzer (Model Epsilon 5, Almelo, The Netherlands) (Watson et al., 1999). The
backup KOH-impregnated cellelose-fiber filters behind the Teflon-membrane front filter were analyzed for acidic gases,
including hydrogen fluoride (HF), hydrochloric acid (HCl), nitrous acid (HNO$_2$), nitric acid (HNO$_3$), and sufuric acid (H$_2$SO$_4$)
as their corresponding ions (i.e., F$^-$, Cl$^-$, NO$_2^-$, NO$_3^-$, and SO$_4^{2-}$) by ion chromatography (IC) (Sturges and Harrison, 1989;
Eldering et al., 1991), which are known to emit from waste burning (Christian et al., 2010).

Channel 2 contains a quartz-fiber filter backed by a citric acid-impregnated cellelose-fiber filter. Half of the quartz-fiber
filter was extracted in DDW and analyzed for ten water-soluble ions, including: ammonium (NH$_4^+$), sodium (Na$^+$), magnesium
(Mg$^{2+}$), potassium (K$^+$), calcium (Ca$^{2+}$), fluoride (F$^-$), chloride (Cl$^-$), nitrite (NO$_2^-$), nitrate (NO$_3^-$), and sulfate (SO$_4^{2-}$) by using
Dionex ICS 6000 IC systems (Thermo Scientific, Sunnyvale, CA) (Chow and Watson, 2017). The backup citric acid-
impregnated cellelose-fiber filter behind the quartz-fiber front filter was anlayzed for ammonia (NH$_3$) as NH$_4^+$ by IC.

Organic and elemental carbon (OC and EC), and eight thermal fractions (i.e., OC1–OC4, pyrolyzed carbon [OP], EC1–
EC3) were quantified following the IMPROVE_A thermal/optical protocol using the DRI Model 2015 Multiwavelength
Carbon Analyzer (Magee Scientific, Berkeley, CA) (Chow et al., 2007; 2015b; Chen et al., 2015). A 0.5 cm$^2$ punch was taken
from the Channel 2 quartz-fiber filter and heated in a pure helium (He) carrier gas at 140 ℃ (OC1), 280 ℃ (OC2), 480 ℃
(OC3), and 580 ℃ (OC4) temperature steps. Next, the carrier gas composition was changed to 98% He/2% O$_2$, and the filter
continued to be heated at 580 ℃ (EC1), 740 ℃ (EC2), and 840 ℃ (EC3). Seven lasers with wavelengths ranging from 405
nm to 980 nm were used to monitor light reflectance (R) and transmittance (T), which were used to calculate wavelength
dependent light absorption. OC and EC were determined at the 635 nm wavelength after R returned to its initial value to correct
for OC pyrolysis. The multiwavelength measurement allowed separation of light absorption by black carbon (BC) from brown
carbon (BrC), which has unique wavelength dependence based on fuel and combustion conditions (Chow et al., 2015b; 2018;
2021).

A parallel quartz-fiber filter in Channel 3 was analyzed for non-polar organic compounds, including PAHs, nitro-PAHs,
alkanes, cycloalkanes, hopanes, steranes, phthalates, and other organics using in-injection port-thermal desorption-gas
chromatography mass spectrometry (TD-GC/MS) (Ho et al., 2008; Ho and Yu, 2004; Ho et al., 2011). Aliquots (1.0–1.5 cm$^2$)
of the quartz-fiber filters were cut into small pieces, spiked with internal standards, and inserted into TD tubes for analyses.

Chemical data were quality checked as part of quality assurance (QA) to ensure internal consistency and to achieve mass
closure. As shown in Supplemental Figure S1, the sum of measured chemical species and reconstructed mass accounted for
73% and 99% of gravimetric mass on average, respectively. High coefficients of determination (R$^2$) of 0.98 assure that major
PM$_{2.5}$ constituents (i.e., gravimetric mass, carbon, ions, and elements) are quantified with high quality (Chow et al., 2015a).
To obtain chemical source profiles, the chemical concentrations were normalized to PM$_{2.5}$ mass concentrations. Potential
markers and hazardous air pollutants emissions from each waste category were assessed. Fuel-based EFs were calculated based
on carbon mass balance techniques (Wang et al., 2023).





## 3 Results and Discussion

### 3.1 Major PM$_{2.5}$ Compositions

Figure 2 compares average PM$_{2.5}$ mass fractions for the five major composition categories (i.e., organic matter OM, EC, ions, mineral, and others). Carbonaceous aerosol (OM and EC) contributed more than 70% of PM$_{2.5}$ mass with minor

contributions from ions. OM was the most abundant component, accounting for >50% of PM$_{2.5}$ mass. EC had the highest abundances in the flaming materials, including plastic bags (49.4±29.2%), combined fuels (47.9±13.2%), textile (12.8±4.3%), and dry vegetations (9.1±2.2%). Detailed mass fractions for each material are shown in Figure S2. High OC and EC abundances were also found for PM$_{2.5}$ from burning of mixed waste and plastics in Nepal (Jayarathne et al., 2018) and China (Wu et al., 2021; Cheng et al., 2020).

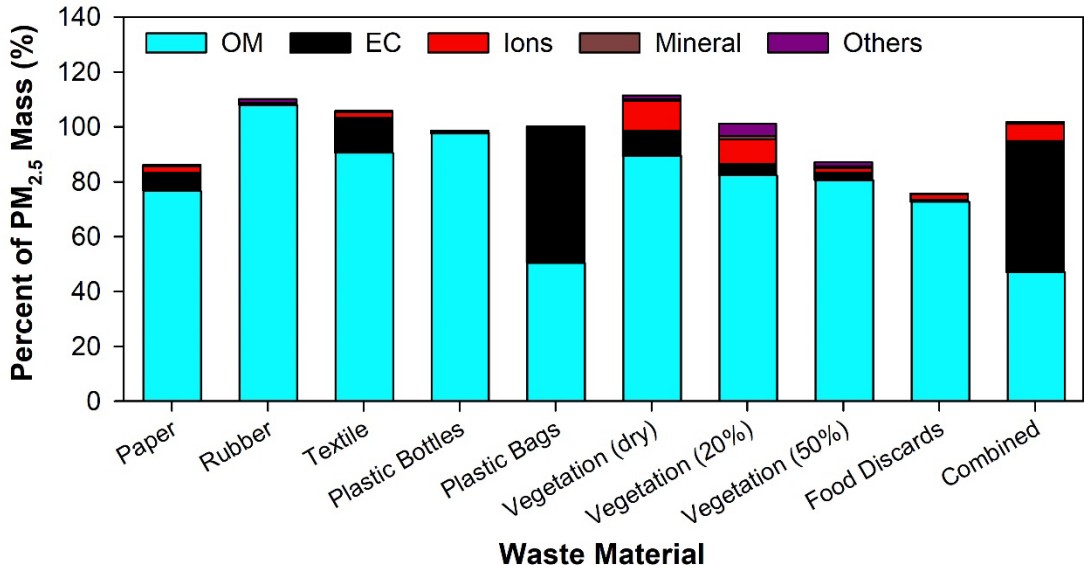


**Figure 2: Abundances of major PM$_{2.5}$ compositions (percent of PM$_{2.5}$ mass) from burning of different waste materials (organic matter= OC × 1.4; ions is the sum of ammonium (NH$_4^+$), sodium (Na$^+$), magnesium (Mg$^{2+}$), potassium (K$^+$), calcium (Ca$^{2+}$), fluoride (F$^-$), chloride (Cl$^-$), nitrite (NO$_2^-$), nitrate (NO$_3^-$), and sulfate (SO$_4^{2-}$) by IC; minerals = 2.2×Al + 2.49×Si + 1.63×Ca + 2.42×Fe + 1.94×Ti.**

The abundances of the seven carbon fractions (i.e., OC1–OC4 and EC1–EC3) by source type are shown in Figure S3. The sum of lower temperature OC1 and OC2 fractions exceeded 20% for most fuels, except for plastic bags and combined materials that had intense flaming combustion. Paper and vegetation had higher OC3+OC4 fractions (15–30%), consistent with their higher charring fractions (pyrolysis of OC to EC during oxygen-free heating). Plastic bags and combined materials had the highest sums of EC1 and EC2 (53–70%), with plastic bags having much higher EC2 (63%) than EC1 (8%), indicating higher

combustion temperatures. High temperature EC3 (840ºC) were not detected. Carbon fractions varied by moisture content in vegetation samples with the highest OC3, OC4, and EC1 in dry vegetation due to dominant flaming combustion. As moisture content increased to 50%, the abundances of OC1 increased, OC2 remained approximately the same, while OC3, OC4, and



EC1 abundances decreased. Similar carbon fraction distributions were found for barrel and pile burning of MSW by Cheng et al. (2020).

Figure 3 shows EFs for $PM_{2.5}$ and its major components OM and EC. Rubber, plastic bottles, 50% moisture vegetation, and food discards generated higher EFs for $PM_{2.5}$ and OM among 10 source types, consistent with their dominant smoldering combustion. Combustion of plastic bottles produced the highest EFs for $PM_{2.5}$ (651 ± 38 g kg$^{-1}$) and OM (635 ± 49 g kg$^{-1}$).

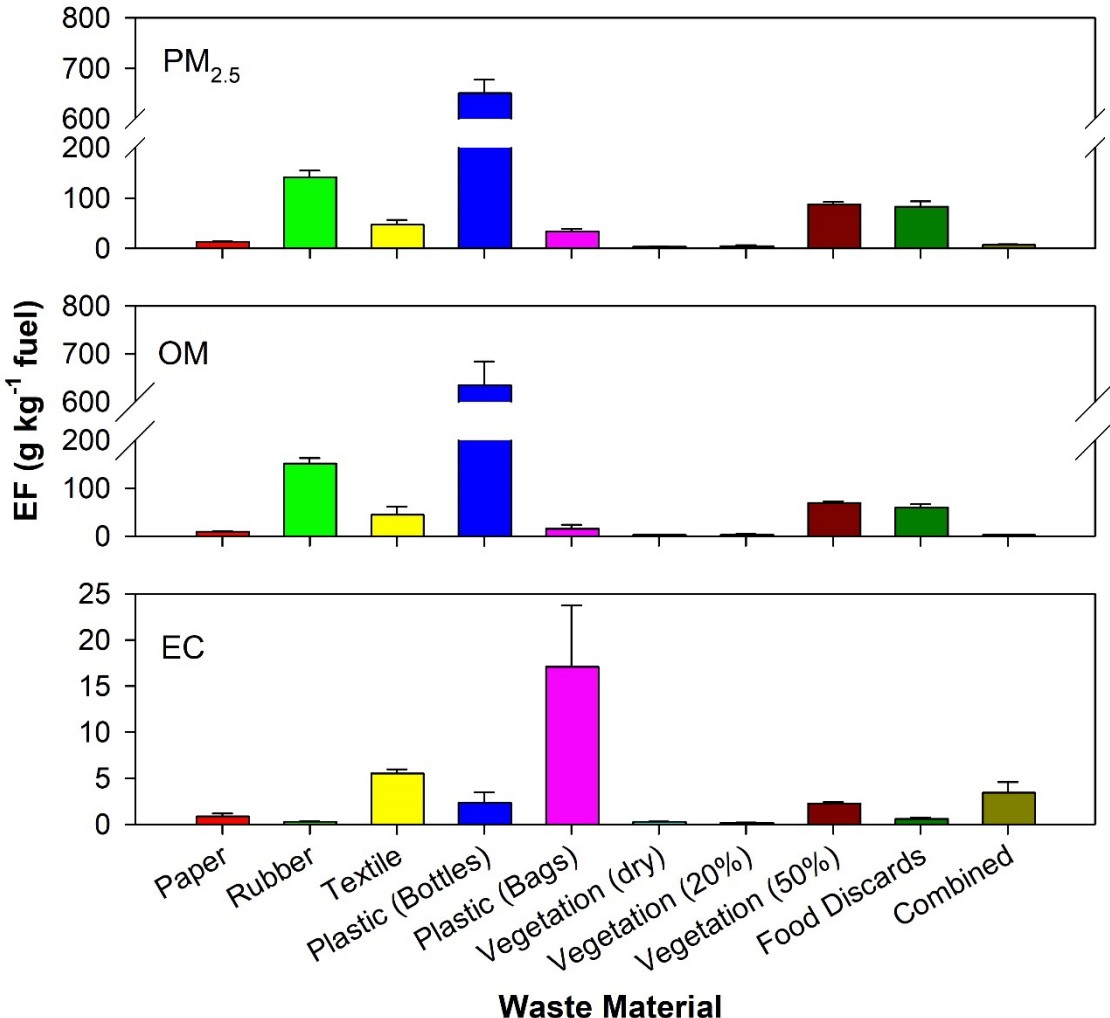

**Figure 3: Emission factors for PM$_{2.5}$, organic matter (OM= OC × 1.4), and elemental carbon (EC). The error bars represent the**
**larger of the propagated analytical uncertainty or the standard error of multiple runs.**

As shown in Figure S2d, over 97% of the plastic bottle $PM_{2.5}$ is OM. Rubber had the second highest EFs for $PM_{2.5}$ (141 ± 23 g kg$^{-1}$) with 98% $PM_{2.5}$ mass being OM (Figure S2b). Plastic bags had the highest EFs for EC (17.1 ± 1.9 g kg$^{-1}$), followed by textile (5.52 ± 0.24 g kg$^{-1}$) and combined waste materials (3.44 ± 1.98 g kg$^{-1}$). Assuming a 100-year global warming potential



(GWP) of 900 for EC (Bond et al., 2013), the climate forcing of EC from these materials is a factor of 2–5 higher than that of
$CO_2$ (Wang et al., 2023).

As 50% moisture vegetation only smoldered, the $PM_{2.5}$ EF (88 ± 7 g kg⁻¹) was over an order of magnitude higher than those of dry (3.2 ± 1.3 g kg⁻¹) and 20% moisture (4.8 ± 2.0 g kg⁻¹) samples. Even though the dry vegetation had a higher EC abundance than the 50% moisture vegetation (8.2% vs. 2.6% of $PM_{2.5}$), the 50% moisture vegetation still had a higher EF for EC due to the much higher EF for $PM_{2.5}$ (Figure 3).

As thermal EC and optical BC are often used interchangeably (Pöschl, 2003), Table 1 compares EC from this study with EC or BC reported in the literature. The EC EF for paper (0.86 ± 0.57 g kg⁻¹) is in the same range as those (0.5–0.76 g kg⁻¹) reported by Cheng et al. (2020) and Wu et al. (2021), while the EF for textile is about 4 times those reported by Cheng et al. (2020). A wide range of EFs are reported for plastic bottles and bags. The plastic bag EF (17.1 ± 11.6 g kg⁻¹) is in good agreement with plastic foam (18.7± 3.9 g kg⁻¹) by Wu et al. (2021) and close to a plastic mixture (10.3 ± 1.0 g kg⁻¹) burned by
Jayarathne et al. (2018). Different vegetation types, moisture contents, and burning conditions resulted in variable EFs. However, EFs for this study fall in the range reported by Akagi et al. (2011) except for 50% moisture vegetation that is 75% higher than the maximum value by Akagi et al. (2011). The EF for mixed materials (3.44± 1.98 g kg⁻¹) is similar to those reported for damp mixed garbage (3.30 ± 3.88 g kg⁻¹) in Nepal (Jayarathne et al., 2018; Stockwell et al., 2016) but is five times the suggested value (0.65 ± 0.27 g kg⁻¹)  for global emission inventory (Akagi et al., 2011; Wiedinmyer et al., 2014).


**Table 1: Comparison of EC or BC and PAH emission factors measured in this study with those from the literature.**

| Ref. | Region | Fuel | Method | EC or BC (g kg⁻¹ fuel) | PAHs (g kg⁻¹ fuel) |
|---|---|---|---|---|---|
| **Paper** | | | | | |
| **This study** | **South Africa** | **Paper** | **Lab burning** | **0.86 ± 0.57** | **0.051 ± 0.001** |
| (Park et al., 2013) | South Korea | Paper | Lab incinerator | | 0.002 |
| (Hoffer et al., 2020) | Hungary | Advertising flyer and newspaper | Lab stove | | 0.0012 ± 0.0008 |
| (Cheng et al., 2020) | China | Paper | Lab barrel | 0.76 ±0.01 | |
| (Cheng et al., 2020) | China | Paper | Lab pile | 0.50 ±0.02 | |
| (Wu et al., 2021) | China | Paper packaging | Field | 0.50 ± 0.11 | 0.031 ± 0.018 |
| **Leather/Rubber** | | | | | |
| **This study** | **South Africa** | **Synthetic car floor mat** | **Lab burning** | **0.29 ± 0.08** | **1.41 ± 0.06** |
| (Hoffer et al., 2020) | Hungary | Tire | Lab stove | | 0.025 ± 0.009 |
| **Textile** | | | | | |



| Ref. | Region | Fuel | Method | EC or BC (g kg⁻¹ fuel) | PAHs (g kg⁻¹ fuel) |
|---|---|---|---|---|---|
| **This study** | **South Africa** | **Cloth** | **Lab burning** | **5.52 ± 0.24** | **0.015 ± 0.005** |
| (Hoffer et al., 2020) | Hungary | Cloth | Lab stove | | 0.021 ± 0.019 |
| (Cheng et al., 2020) | China | Textile | Lab barrel | 1.47 ± 0.13 | |
| (Cheng et al., 2020) | China | Textile | Lab pile | 1.30 ±0.09 | |
| **Plastics** | | | | | |
| **This study** | **South Africa** | **Plastic bottles** | **Lab burning** | **2.37 ± 1.89** | **8.55 ± 2.02** |
| **This study** | **South Africa** | **Plastic bags** | **Lab burning** | **17.1 ± 11.6** | **0.24 ± 0.04** |
| (Park et al., 2013) | South Korea | Plastics | Lab incinerator | | 0.007 |
| (Jayarathne et al., 2018) | Nepal | Chip bags (damp) | Field | 5.71 ± 0.58 | 0.076 |
| (Jayarathne et al., 2018) | Nepal | Plastics (mostly heavy clear plastic, some plastic cups, and food bags) | Field | 10.31 ± 1.04 | 0.152 |
| (Hoffer et al., 2020) | Hungary | Different types of plastics | Lab stove | | 0.03–0.26 |
| (Wu et al., 2021) | China | Plastic packaging | Field | 0.22–0.70 | 0.017–0.03 |
| (Wu et al., 2021) | China | Plastic foam | Field | 18.7 ± 3.9 | 0.256 ± 0.093 |
| **Vegetation** | | | | | |
| **This study** | **South Africa** | **Vegetation (0% H₂O)** | **Lab burning** | **0.27 ± 0.03** | **0.010 ± 0.004** |
| **This study** | **South Africa** | **Vegetation (20% H₂O)** | **Lab burning** | **0.19 ± 0.04** | **0.001±0.001** |
| **This study** | **South Africa** | **Vegetation (50% H₂O)** | **Lab burning** | **2.28 ± 0.21** | **0.046 ± 0.003** |
| **This study** | **South Africa** | **Food discards** | **Lab burning** | **0.60 ± 0.25** | **0.173 ± 0.069** |
| (Christian et al., 2010) | Mexico | Cooking biofuels | Field/Lab | 0.205–0.674 | |
| (Akagi et al., 2011) | Worldwide | Biomass | Data Synthesis | 0.2–1.3 | |
| (Park et al., 2013) | South Korea | Wood | Lab incinerator | | 0.001 |
| (Jayarathne et al., 2018) | Nepal | Crop residue (Rice, wheat, mustard, lentil, & weed grass) | Field | 0.98 ± 0.12 | 0.011 |
| (Wu et al., 2021) | China | Organic waste | Field | 0.54 ± 0.39 | 0.032 ± 0.014 |
| **Mixed household waste** | | | | | |
| **This study** | **South Africa** | **Mixed garbage** | **Lab burning** | **3.44 ± 1.98** | **0.024 ± 0.010** |
| (Lemieux, 1997) | U.S. | Recycler waste | Lab barrel | | 0.0235–0.0244 |





| Ref. | Region | Fuel | Method | EC or BC (g kg⁻¹ fuel) | PAHs (g kg⁻¹ fuel) |
|---|---|---|---|---|---|
| (Lemieux, 1997) | U.S. | Non-recycler waste | Lab barrel | | 0.0497–0.0824 |
| (Christian et al., 2010) | Mexico | Landfill MSW | Field | 0.381–0.924 | |
| (Akagi et al., 2011; Wiedinmyer et al., 2014) | U.S. and Mexico | Mixed waste | Data Synthesis | 0.65 ± 0.27 | |
| (Park et al., 2013) | South Korea | Domestic municipal solid waste | Lab incinerator | | 0.0015 |
| (Stockwell et al., 2016) | Nepal | Mixed waste | Field | 3.30 ± 3.88 | |
| (Jayarathne et al., 2018) | Nepal | Dry mixed garbage | Field | <0.04 | 0.015 |
| (Jayarathne et al., 2018) | Nepal | Damp mixed garbage | Field | 1.56–3.41 | 0.097-0.149 |
| (Cheng et al., 2020) | China | Mixed waste | Lab barrel | 1.26 ± 0.16 | |
| (Cheng et al., 2020) | China | Mixed waste | Lab pile | 1.03 ±0.13 | |

## 3.2 Elements

Figure 4 shows EFs for elements measured by XRF with EF values larger than uncertainties for at least three of the ten
waste materials or those in the Hazardous Air Pollutants (HAPs) list of the U.S. Environmental Protection Agency (U.S. EPA,
2020). Chlorine (Cl) had the highest EFs for all waste materials. Rubber had the highest EFs for Cl and sulfur (S) as well as
HAP elements cadmium (Cd), antimony (Sb), and lead (Pb), while the 50% moist vegetation had the highest EFs for chromium
(Cr), cobalt (Co), nickel (Ni), and selenium (Se). Table S1 compares these heavy metal EFs with those reported in the literature.
The EFs by Park et al. (2013) are lower than those by other studies for most elements, except for higher zinc (Zn) EFs. For
paper burning, both this study and Cheng et al. (2020) found similar EFs for copper (Cu) and Pb. The plastic bottles had much
higher EFs for Cr and Pb than plastic bags and other studies; high Cu EFs for plastics are also found from this study and Cheng
et al. (2020). The high Cu emissions from paper and plastics likely originate from the Cu compounds used for cyan, green and
reddish blue printing pigment (Zięba-Palus and Trzcińska, 2011). The EFs for Cr, Zn, and Pb from dry and 20% moisture
vegetations are similar to those by Cheng et al. (2020), while the Cu EFs for 50% moisture vegetation is similar to that by
Cheng et al. (2020). For the combined materials, Cr and Ni are below detection limits in this study, but the EFs for other
elements are in the same range as those reported in the literature, with large variations (Lemieux, 1997; Christian et al., 2010;
Jayarathne et al., 2018; Cheng et al., 2020).



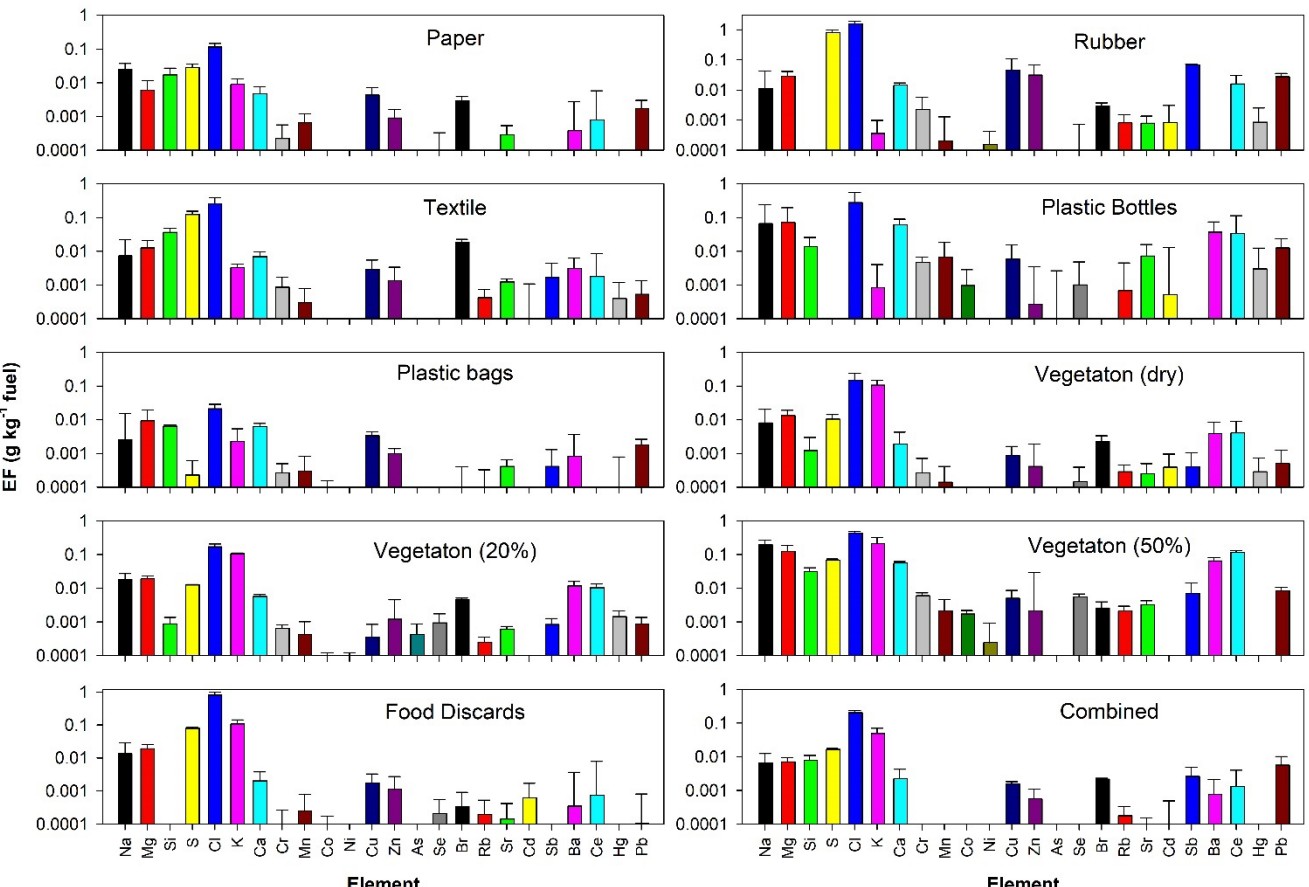

**Figure 4: Emission factors for key elements.**

The elemental abundances are shown in Figure S4. Cl abundance ranged from 0.04±0.04% of $PM_{2.5}$ mass for plastic bags to 5.1±4.3% for dry vegetation. The low Cl abundance in the tested plastics was probably due to their lack of polyvinyl chloride (PVC) components. Vegetation samples had higher potassium (K) abundances, consistent with the fact that K is often used as a biomass burning marker. Plastic bottles and bags had lower abundances of most elements than other materials, similar to the findings by Valavanidis et al. (2008). Sb has been reported as a potential marker for solid waste burning because of its use in

textile (flame retardant), batteries, and polyethylene terephthalate (PET) projection (Jayarathne et al., 2018; Christian et al., 2010). Figure S4 shows that Sb had higher abundances in rubber, vegetation, and mixed materials, but with lower abundances in paper, plastics, and food discards, indicating that caution should be taken when using Sb as a waste burning marker due to its high dependence on waste compositions. The Sb abundance in the combined materials (0.035±0.021%) is similar to the value of 0.025±0.033% reported by Jayarathne et al. (2018). The combined materials had higher abundances of Pb

(0.075±0.062%) than other waste categories, probably related to the metals and glass added to the mix; this value is similar to 0.057±0.077% reported by Jayarathne et al. (2018).



### 3.3 Acidic and Alkali Gases and Ions

Acidic and alkali ionic species are present in both gas and particle phases and their partition depends on temperature and reactions. Acidic and alkali gases are toxic and corrosive; they can cause adverse effects on human health, materials, and ecosystems if not neutralized soon after emission. Figure S5 shows elevated ionic concentrations for $NO_3^-$ and $NH_4^+$ on backup filters (i.e., in the form of $HNO_3$ and $NH_3$, respectively). The sum of gaseous ion abundances ranged from 0.6% (plastic bottles) to 73% (dry vegetation) of $PM_{2.5}$, higher than the particulate ions abundances (less than 10% of $PM_{2.5}$). Figure S6 shows abundant $Cl^-$ and $SO_4^{2-}$ for dry and 20% vegetation. Consistent with high abundances of elemental K, dry and 20% moisture vegetation show high $K^+$ abundances. Vegetation with 50% moisture content had low $K^+$ abundances, probably because the dominant smoldering phase left most K in the ash. The sums of anion equivalents were higher than those for cations for most materials, likely because hydrogen ion ($H^+$) was not measured; it might be associated with hydrochloric, sulfuric, or nitric acids. Therefore, directly emitted particles appear to be acidic, although these would probably be neutralized by available $NH_3$ soon after emission. The deposition of the acidic waste burning particles probably contributed to discoloring of the Taj Mahal in India (Lal et al., 2016).

Figures 5 and 6 show that EFs for HCl from rubber; $HNO_3$ from rubber, textile, vegetation, and food discards; and $NH_3$ from food discards were one to two orders of magnitude higher than the corresponding particulate ions. Food discards and 50% moisture vegetation had the highest EFs for HF and particulate $F^-$, consistent with fluorine accumulation in plants (Jayarathne et al., 2014). One would expect that the dry and 20% moisture vegetation samples would cause higher EFs for HF than particulate $F^-$ due to higher combustion temperatures. However, low HF and $F^-$ EFs were observed in these cases. Future study should investigate the partitioning of fluorine among gases, particles, and ashes during biomass burning. Jayarathne et al. (2014) reported $F^-$ EFs in the range of 0.7–136 mg kg$^{-1}$ for several types of biomass burning with an overall average of 32 ± 7 mg kg$^{-1}$. These values are in the range of dry (7.6 ± 0.6 mg kg$^{-1}$) and 25% moisture vegetations (17.8 ± 8.3 mg kg$^{-1}$), but lower than the 50% moisture vegetation (744 ± 61 mg kg$^{-1}$) and food discards (291 ± 88 mg kg$^{-1}$).

The rubber sample had the highest EF for HCl (9.6 ± 1.5 g kg$^{-1}$) with an order of magnitude lower EF for PM $Cl^-$ (0.8 ± 0.2 g kg$^{-1}$). Consistent with low elemental Cl emission, the plastic bottles or bags did not have high HCl or $Cl^-$ emissions. Lemieux (1997) reported HCl EFs of 1.51–3.28 and 0.086–0.481 g kg$^{-1}$ for combined waste materials with higher and lower PVC mass fractions, respectively. The EF for lower PVC waste is similar to the values for combined materials (0.47 ± 0.22 g kg$^{-1}$) measured in this study. These values are lower than the 1.7–9.8 g kg$^{-1}$ EFs reported by Christian et al. (2010) for landfill MSW burning in Mexico. Stockwell et al. (2016) measured HCl from six mixed garbage samples and found EFs ranging from non-detectable to 3.03 kg$^{-1}$. They also found that one sample containing mostly plastic bags did not have detectable HCl but another sample dominated by hard plastics had a high HCl EF of 77.9 g kg$^{-1}$, indicating the high sensitivity of HCl EFs to fuel compositions.



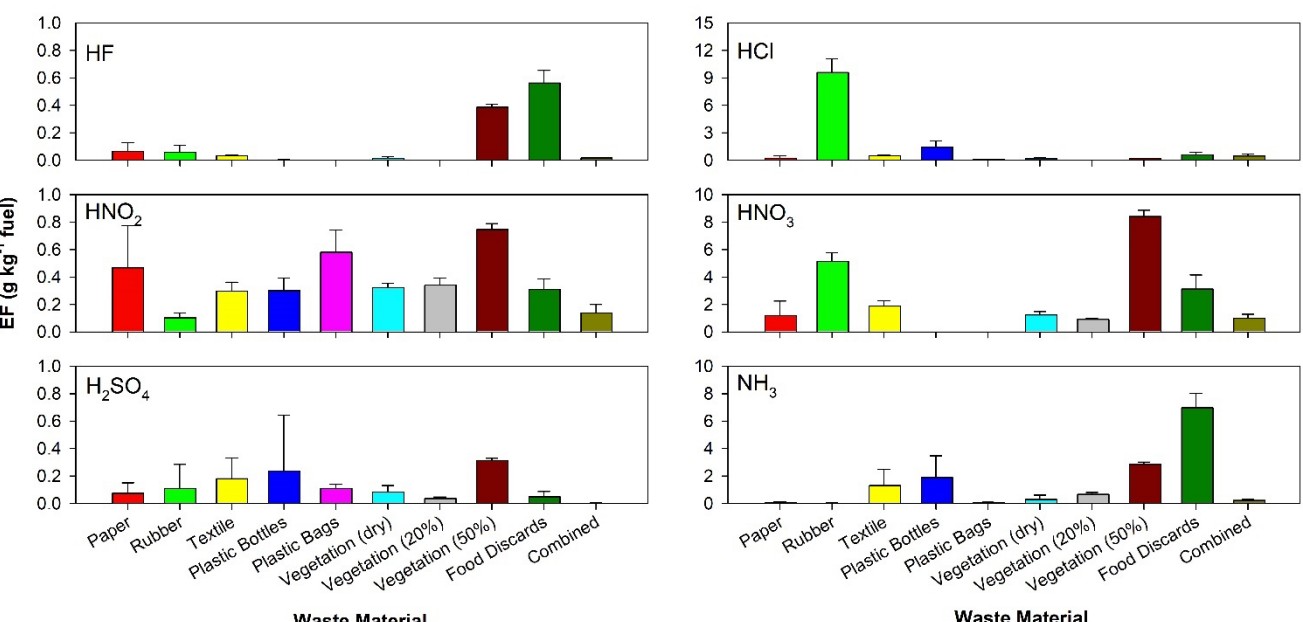

**Figure 5: Emission factors for acidic and alkali gases.**

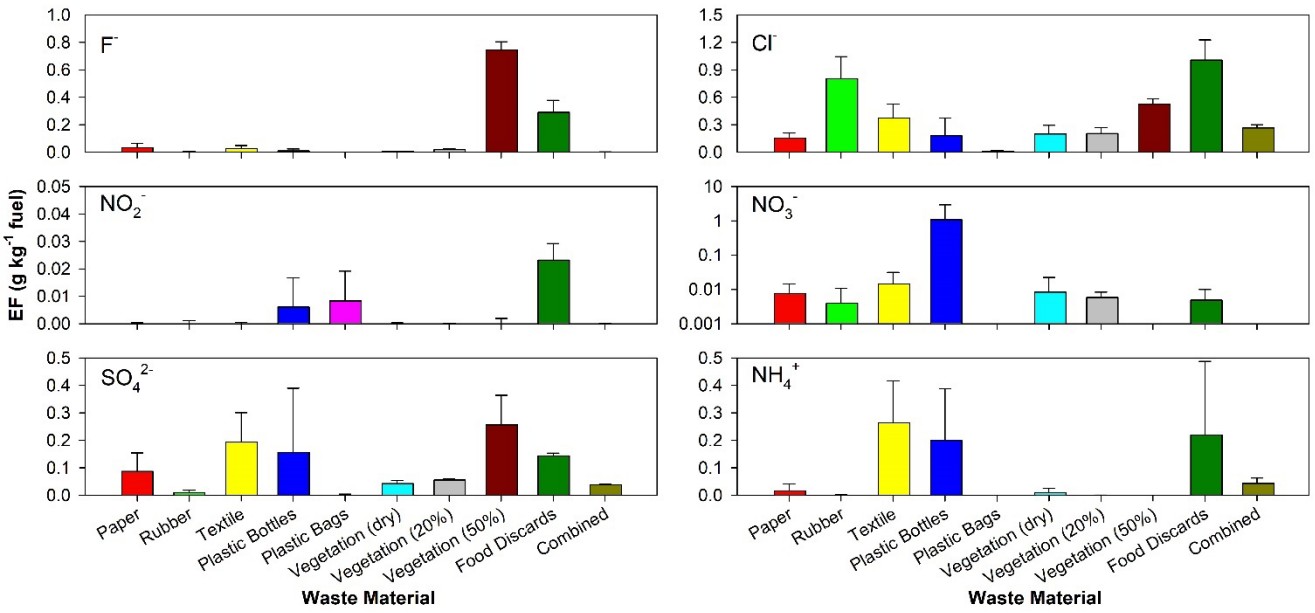

**Figure 6: Emission factors for ionic particulate species corresponding to the acidic and alkali gases.**

Due to high volatilities, $NO_2^-$ and $NO_3^-$ were in the gas phase (Figure S5). The waste materials had low sulfur contents (Wang et al., 2023); therefore, EFs for both $H_2SO_4$ and $SO_4^{2-}$ were low. $NH_3$ was in gas phase because the fresh emissions had not reacted with acidic gases to create PM ammonium. Similar to HF emissions, 50% moisture vegetation ($2.86 \pm 0.16$ g kg$^{-1}$) and food discards ($6.98 \pm 1.05$ g kg$^{-1}$) had the highest $NH_3$ EFs. The $NH_3$ EFs for the combined materials was $0.23 \pm 0.08$ g



kg$^{-1}$, within the large variations of 0.94 ± 1.02 g kg$^{-1}$ reported by Akagi et al. (2011). The NH$_3$ EFs for the vegetations (0.29–2.86 g kg$^{-1}$) from this study are similar to the 0.52–2.72 g kg$^{-1}$ found by Akagi et al. (2011) for different biomass burning emissions.

### 3.4 Non-polar Organic Compounds

The abundances (Figure 7) and EFs (Figure S7) for organic groups differ among waste materials. Higher abundances of n-alkanes, phthalate, and PAHs are found for rubber and plastics (Figure 7), consistent with elevated EFs (Figure S8). Paper and textile emissions are dominated by n-alkanes. The high abundance of n-alkanes and phthalates in solid waste burning is consistent with prior studies (Jayarathne et al., 2018; Simoneit et al., 2005).

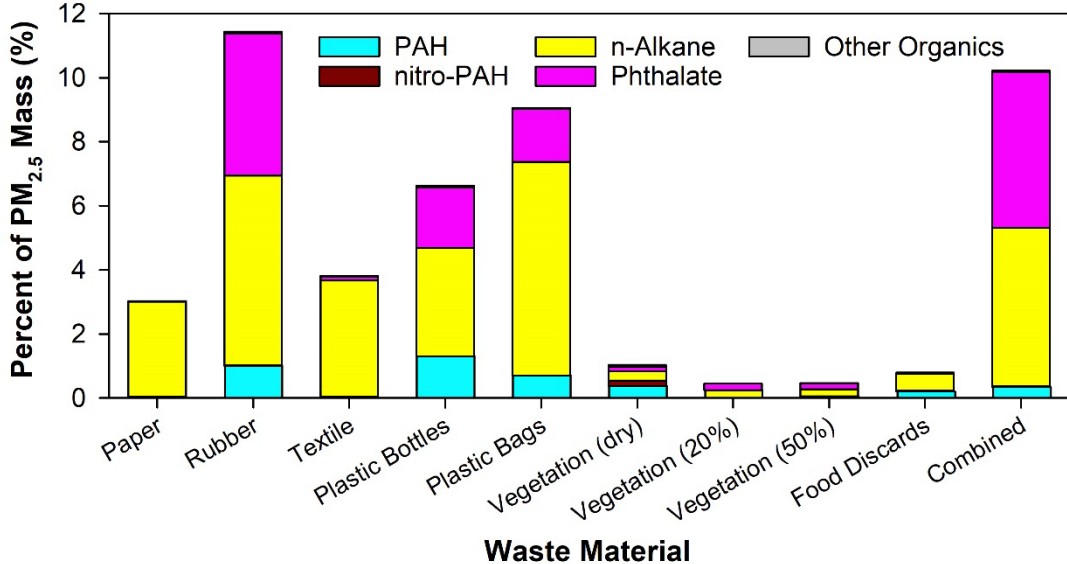

**Figure 7: Abundances for organic groups (% of PM$_{2.5}$ mass).**

Figure 8 shows the EFs for the most abundant PAHs including the U.S. EPA priority PAHs (Andersson and Achten, 2015). The PAH distribution shows similar patterns for rubber and plastic bottles/bags with dominant EFs and abundances (Figure S8) found for 5- and 6-ring PAHs (e.g., benzo[b]fluoranthene, benzo[j+k]fluoranthene, and indeno[1,2,3-cd]pyrene), although the relative abundances for rubber differ from those for plastic bottles and bags. Simoneit et al. (2005) also reported

that benzo[j+k]fluoranthene (0.02–0.036% of PM$_{10}$ mass) and indeno[1,2,3-cd]pyrene (0.01–0.024% of PM$_{10}$ mass) were the most abundant PAHs from plastics combustion. However, their abundances were an order of magnitude lower than the abundances found in this study (0.1–0.3% of PM$_{2.5}$ mass for both PAH species). Paper and textiles consist mostly of 4-ring PAHs, with high EFs for fluoranthene, pyrene, benzo[a]anthracene, and benzo[c]phenanthrene. The PAH EFs for textile were about twice those for paper. Vegetation and food discards burning emitted a wide range of PAHs, and the EFs varied

significantly with moisture content.





**Figure 8: Emission factors for PAHs.**

PAH EFs from this study and those reported in the literature are compared in Table 1. Note that PAH compounds measured in different studies may be different. The PAH EF for paper from this study ($0.051 \pm 0.001$ g kg$^{-1}$) is in the same range as that reported by Wu et al. (2021), but is over an order of magnitude higher than those by Park et al. (2013) and Hoffer et al. (2020). Hoffer et al. (2020) also reported much lower PAH EFs for rubber but similar EFs for textile as compared to this study. The plastic bottle PAH EF from this study is over an order of magnitude higher than the other studies while the plastic bag and vegetation PAH EFs from this study are consistent with several other studies (Jayarathne et al., 2018; Hoffer et al., 2020; Wu et al., 2021). However, the EFs for food discards are over 10 times higher than those for vegetation. The combined waste PAH EF from this study ($0.024 \pm 0.010$ g kg$^{-1}$) is close to that for recycled waste ($0.0235$–$0.0244$ g kg$^{-1}$) by Lemieux (1997) and





dry mixed waste (0.015 g kg$^{-1}$) by Jayarathne et al. (2018). These EFs are lower than those reported for non-recycled waste (Lemieux, 1997) and damp mixed waste (Jayarathne et al., 2018).

PAH diagnostic ratios have been used to infer PAH sources (Tobiszewski and Namieśnik, 2012; Ravindra et al., 2008; Harner et al., 2018); however, not many ratios have been reported for MSW burning emissions (James et al., 2023). Table S2

lists several common PAH diagnostic ratios from this study. While there are similarities among different materials, some distinct ratios can be observed. For example, plastic bottles and bags have significantly (p<0.05) higher benzo[e]pyrene/benzo[a]pyrene and indeno[1,2,3-c,d]pyrene/benzo[g,h,i]perylene ratios; however, they had much lower emissions of retene, which is a biomass burning emission tracer, than other materials.

Figure 9 and Figure S9 show that 2-nitrobuphenyl is the most abundant nitro-PAH for most MSW. Plastic bottles had the

highest nitro-PAH EFs, followed by food discards and rubber. The dry vegetation had higher EFs than the moisture fuels.

**Figure 9: Emission factors for nitro-PAHs.**



The cancer risks of PAHs and nitro-PAHs are often estimated using the equivalents of benzo(a)pyrene (BaP$_{eq}$), one of the most potent PAHs with known carcinogenic characteristics. The toxic equivalent factor (TEF) for BaP is set to 1 and other
PAHs and nitro-PAHs are assigned a TEF value by comparing their relative toxicity to that of BaP (ATSDR, 2022; Samburova et al., 2017; Moradi et al., 2022). For PAHs, dibenzo[a,h]anthracene has the highest TEF of 2.4, followed by BaP, benzo[e]pyrene, and dibenzo[a,e]pyrene with a TEF of 1.0 (ATSDR, 2022; Samburova et al., 2017). For nitro-PAHs, 6-nitrochrysene and 1,6-dinitropyrene have the highest TEF of 10 (ATSDR, 2022). The EFs for BaP$_{eq}$ (BaP toxicity equivalent) were calculated from the sums of the products of the EF and TEF of individual PAHs and nitro-PAHs. Figure 10 shows that
plastic bottles, rubber, and plastic bags had the highest PAH EFs for BaP$_{eq}$, while plastic bottles, food discards, and rubber had the highest nitro-PAH EFs for BaP$_{eq}$, consistent with their high EFs for PAHs and nitro-PAHs. The PAH BaP$_{eq}$ EFs for paper (0.00032 ± 0.00010 g kg$^{-1}$) and textile (0.0010 ± 0.0003 g kg$^{-1}$) are in similar ranges with those reported by Hoffer et al. (2020): 0.00016 ± 0.00012 g kg$^{-1}$ for paper and 0.0016 ± 0.0020 g kg$^{-1}$ for textile. However, the PAH BaP$_{eq}$ EFs for rubber (0.47 ± 0.06 g kg$^{-1}$), plastic bags (0.47 ± 0.06 g kg$^{-1}$), and plastic bottles (3.0 ± 0.4 g kg$^{-1}$) were over an order of magnitude higher than
the tire and plastics (0.001–0.02 g kg$^{-1}$) reported by Hoffer et al. (2020). These differences are likely caused by the more efficient oxidation by co-combustion of solid waste with charcoal in the Hoffer et al. (2020) tests and more PAH species were measured and included in the BaP$_{eq}$ calculation in this study.

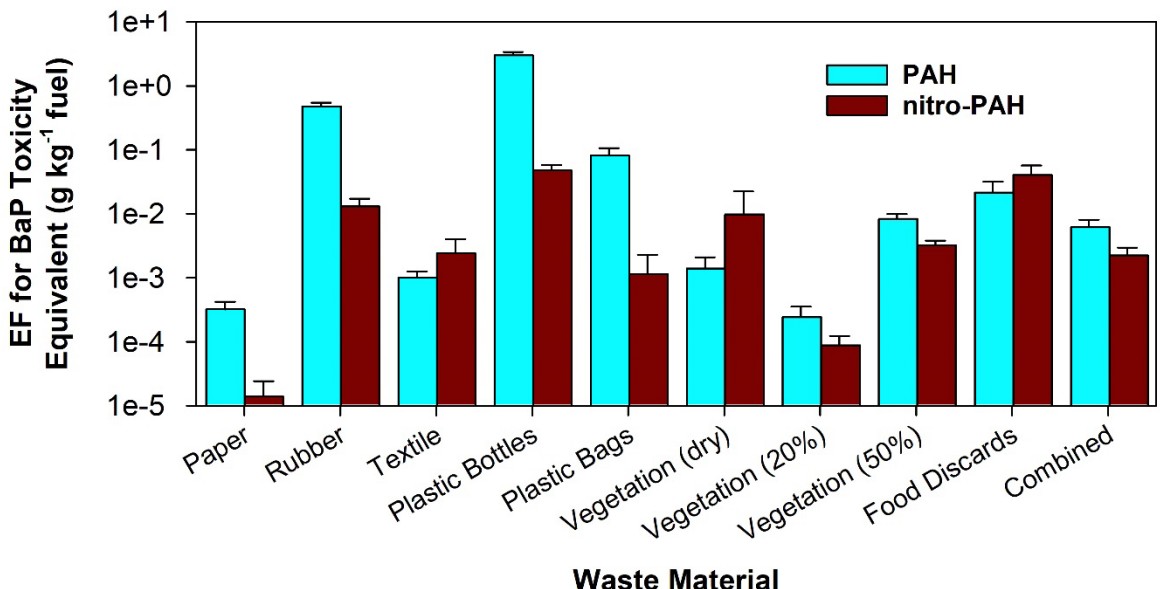

**Figure 10: Emission factors for total PAHs and nitro-PAHs expressed in benzo[a]pyrene (BaP) toxicity equivalent.**

Abundances and EFs for n-alkanes are shown in Figure S10 and Fig. 11, respectively. The most striking preference of odd alkanes over even alkanes was observed for the dry vegetation emissions, with a carbon preference index (CPI; the ratio of the sums of odd to even carbon numbers) of 6.01 ± 0.47, consistent with literature findings (Rogge et al., 1993; Noblet et al., 2021;




Caumo et al., 2020). The higher than unity CPI is due to biogenic wax emissions. The EFs for 50% moisture vegetation were over an order of magnitude higher than those of dry and 20% moisture vegetation samples. However, the odd n-alkane
preference was lost in emissions from moist vegetation, with CPI values of 0.63 ± 0.01 and 0.75 ± 0.03 for vegetation with 20% and 50% moisture contents, respectively. Paper, textile, and food discard emissions also showed preference for odd n-alkanes, with CPIs of 1.80 ± 0.10, 2.39 ± 0.04 and 2.53 ± 0.21, respectively. Plastic bottles had the highest EFs for n-alkanes, with C30-C32 having the highest EFs. Synthetic rubber and plastic bags had the second and third highest EFs for n-alkanes.

**Figure 11: Emission factors for n-alkanes.**

The CPIs for plastic bottles and bags, rubber, and combined waste material were less than 0.6, indicating combustion of petroleum products (Rogge et al., 1993). Jayarathne et al. (2018) reported CPIs of 0.6–1.1 for mixed waste, similar to the combined materials (0.58 ± 0.01) in this study. Simoneit et al. (2005) reported strong preferences for even carbon number n-





alkanes (CPI = 0.1–0.47) for plastic extracts, but the preference decreased (CPI = 0.68–0.98) in the plastic burning smoke due

to thermal cracking. These values are slightly higher but close to those for plastic bottles (0.58 ± 0.03) and plastic bags (0.53 ± 0.02). The carbon number maxima ($C_{max}$) are C30 and C32 for plastic bottles and C32 and C34 for plastic bags, consistent with the values reported by Simoneit et al. (2005).

Phthalates can irreversibly disrupt the endocrine system, metabolism, and multiple organs (Wang and Qian, 2021; Simoneit et al., 2005). Bis(2-ethylhexyl)phthalate (DEHP), one of the most common phthalates, was designated by the U.S.

Environmental Protection Agency (EPA) as a probable human carcinogen (Miao et al., 2017). EFs (Fig. 12) and percent $PM_{2.5}$ abundance (Figure S11) for phthalates are highest for plastic bottles and bags, rubber, and combined materials, consistent with phthalates' use as plasticizing agents (Chien et al., 2003). Butyl benzyl phthalate (BBP), DEHP, and di-n-octyl phthalate (DnBP) were the phthalate species with the highest EFs for all waste materials. Rubber had higher abundances of DnBP than plastic bottles or bags. Phthalate EFs from 50% moisture vegetations are over an order of magnitude higher than the drier

vegetation samples, while paper had the lowest phthalate EFs.

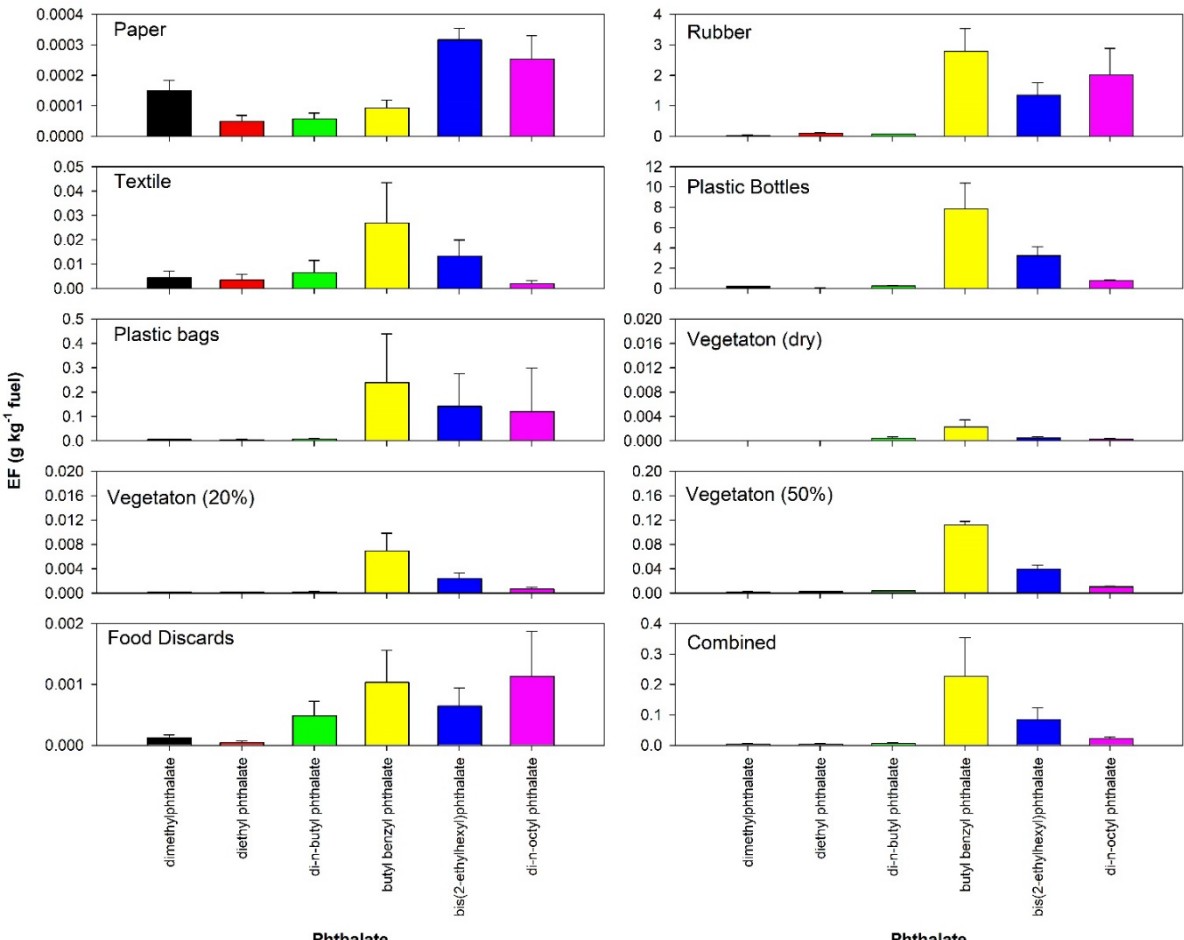

**Figure 12: Emission factors for phthalates.**



### 3.5 Summary of Chemical Characteristics and Emission Factors

Table 2 summarizes the key chemical characteristics of gases and PM$_{2.5}$ emitted from waste burning. These features can
serve as signatures of emission sources in source apportionment studies. OM was the most abundant component (>50% of
PM$_{2.5}$ mass) for all waste materials, and EC was more abundant in the flaming fuels, including plastic bags, combined fuels,
textile, and dry vegetation. The sum of the lower-temperature OC1 and OC2 components exceeded 20% for most fuels, except
for plastic bags and combined fuels which had the highest sums of EC1 and EC2 (53–70%). Cl was the most abundant element
in PM$_{2.5}$ from all waste materials. Vegetation samples had higher abundances of K and K$^+$, confirming their use as biomass
burning markers. Paper and textile non-polar organic emissions were dominated by n-alkanes. They had similar PAH
distributions with abundant 4-ring PAHs (e.g., fluoranthene, pyrene, benzo[a]anthracene, and benzo[c]phenanthrene). Rubber,
plastic bottles, and plastic bag emissions have high abundances of PAHs, n-alkanes, and phthalates, with abundant 5- and 6-
ring PAHs (e.g., benzo[b]fluoranthene, benzo[j+k]fluoranthene, and indeno[1,2,3-cd]pyrene). Dry vegetation emissions
produced approximately 10-fold higher PAH abundances than moist vegetation fuels, with a strong preference for odd n-
alkanes and a CPI of 6.01 ± 0.47. In contrast, rubber, plastic bottles and bags showed preference to even n-alkanes, with CPI
values less than 0.6. Rubber, plastic bottles and bags, and combined materials had the highest abundance of phthalates, while
dry vegetation burning had the highest abundance of nitro-PAHs.

In term of EFs, Cl had the highest EFs among all elements measured by XRF. Rubber had the highest EFs for particulate
Cl and S as well as gaseous HCl due to its most abundant Cl and S in the fuel. Rubber also had the highest EFs for HAP
elements (Cd, Sb, and Pb), while 50% moisture vegetation had the highest EFs for Cr, Co, Ni, and Se. Food discards and 50%
moisture vegetation had the highest EFs for HF, NH$_3$, and particulate F$^-$. The phthalates with the highest EFs are BBP, DEHP,
and DnBP. Plastic bottles had the highest EFs for nitro-PAHs, followed by food discards and rubber. Among all measured
nitro-PAHs, 2-nitrobiphenyl had the highest EFs for most waste materials.

### 4 Conclusions and Discussion

This study provides detailed chemical speciation of filter samples collected from laboratory combustion of ten municipal
solid waste (MSW) materials, representing open burning of household waste in South Africa. Source profiles and emission
factors were calculated. The key conclusions are:

(1) Source profiles representative of local emission sources are critical for accurate source apportionment (Watson et al.,
2016). This study expands conventional elements, ions, and carbon fractions to include non-polar organic compounds
such as PAHs, nitro-PAHs, alkanes and alkenes, and phthalates. The additional chemical speciation allows
improvement in source attributions of open waste burning in South Africa and other regions in the world.





**Table 2: Summary of chemical abundance characteristics of emissions from open burning of different waste materials.**

| Material | Major Composition | Carbon | Elements | Inorganic Ions | Non-polar Organics |
|---|---|---|---|---|---|
| Paper | Flaming, high EC; OM = 76%; EC = 6.5% | High OC3+OC4 = 18.6% | Metals from ink (e.g., Cu) | High $HNO_3$ (8.6%) | Dominated by n-alkanes (3%); similar PAH distribution as textile |
| Rubber | Smoldering, high OM; OM = 98%; EC = 0.2% | High OC1+OC2 = 70% | High Cl (1.1%) and Sb (0.05%) | High HCl (6.7%) | High abundances of PAHs (4.4%), n-alkanes (5.9%), and phthalate (4.4%); high even n-alkane preference (CPI = 0.49) |
| Textile | Flaming, high EC; OM = 86%; EC = 12.1% | High OC1+OC2 = 48% and EC1+EC2 = 20% | | High $NH_4^+$ (0.53%) | Dominated by n-alkanes (3.6%); similar PAH distribution as paper |
| Plastic bottle | Smoldering, high OM; OM = 98%; EC = 0.4% | High OC1+OC2 = 53% | Low elemental abundances | Low gaseous and particulate ions | High abundances of PAHs (1.3%), n-alkanes (3.4%), and phthalate (1.9%); high even n-alkane preference (CPI = 0.58) |
| Plastic bag | Flaming, highest EC; OM = 50%; EC = 49% | Low OC1+OC2 = 7.5% and highest EC2 = 63% | Low elemental abundances | Low gaseous and particulate ions | Highest abundance of n-alkane (6.7%), and high abundances of PAHs (0.7%) and phthalate (1.7%); high even n-alkane preference (CPI = 0.53) |
| Vegetation (dry) | Flaming, high EC and ions; OM = 80%; EC = 8.2%; ions: 9.9% | High OC3+OC4 = 29.6% and EC1 = 15.8% | Highest Cl (5.1%) and K (3.3%) | Highest $HNO_2$ (11%), $HNO_3$ (43%), $Cl^-$ (7.1%) | Low non-polar organic abundances; high odd n-alkane preference (CPI = 6); highest nitro-PAH abundances (0.17%) |
| Vegetation (20%) | Flaming, high ions; OM = 81%; EC = 4.2%; ions: 8.8% | High OC3+OC4 = 22.3% | High Cl (4.0%) and K (2.4%) | Highest $NH_3$ (17%) and $K^+$ (2.1%), and high $HNO_2$ (8%), $HNO_3$ (21%), and $Cl^-$ (5%) | Low non-polar organic abundances |
| Vegetation (50%) | Smoldering, high OM; OM = 81%; EC = 2.6% | High OC1+OC2 = 37% | Lower Cl and K than drier vegetations | Highest $F^-$ (0.85%) | Low non-polar organic abundances |
| Food discards | Smoldering, high OM; OM = 73%; EC = 0.8% | High OC1+OC2 = 38% | Low heavy metals | Similar to 50% moist vegetation | Low non-polar organic abundances |
| Combined | Flaming, high EC and ions; OM = 46%; EC = 47%; ions = 6% | Low OC1+OC2 = 15% and high EC1 + EC2 = 53% | Highest lead abundance (0.08%) | High $HNO_3$ (15%), HCl (6.8%), and $Cl^-$ (4.1%) | High abundances of n-alkanes (5%), and phthalate (4.9%) |



(2) Emission factors derived from the combustion of local materials that represent community-generated solid waste improve the accuracy of emission estimates. Emissions from open burning of MSW is under-studied for local, national, and global emission inventories (Wiedinmyer et al., 2014) with dearth measurements for waste combustion (Rabaji, 2019; Kwatala et al., 2019). This study contributes to the air quality management and research communities by providing experimentally determined EFs for different waste material categories specific to South Africa. These localized EFs can be used to estimate the emission reductions for SASOL's WCI program and to improve MSW open burning emission inventories for South Africa and other countries.

(3) EFs for chemical species provide additional information to assess potential health risks associated with exposure to open burning emissions. The established chemical database can be used for risk assessment to further demonstrate emission reductions for many air toxics (e.g., hazardous gases, heavy metals, and PAHs) beyond criteria pollutants by reducing open waste burning. The EFs for $CO_2$ and EC are also useful for evaluating the climate impacts from MSW open burning.

(4) The EFs determined for nine individual waste categories and the combined waste category offer flexibility in calculating emissions. When the weight composition is known for open burn piles, emissions of chemical species can be calculated by summing those from individual categories; otherwise, the EFs for the combined categories can be used. EFs for $PM_{2.5}$ from this study are within the ranges reported in the literature. This study fills a data gap, particularly for speciated profiles and EFs from burning many waste materials, such as paper, leather/rubber, textiles, and food discards.

**Data availability.** The source profile and emission factor data are available at: Wang, X., 2023, "Data for: Air Pollutant Emissions from Open Burning of Household Solid Waste from South Africa", https://doi.org/10.7910/DVN/QTV9YW, Harvard Dataverse, V1. Additional data is available upon request.

**Author contributions.** XW, JCC, and JGW designed the study; HF conducted the combustion experiments; SSHH conducted organic speciation; XW performed the data analyses and prepared the original paper draft; WC and ASMDV provided waste materials and resources; all authors reviewed and edited the paper.

**Competing interests.** None.

**Financial support.** This research was partially funded by SASOL (grant no. 2000751484) and partially by the Desert Research Institute internal funding.



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
