# Peer review of "Chemically Speciated Air Pollutant Emissions from Open Burning of Household Solid Waste from South Africa"

_EGUsphere, 2023_

## Referee Comment (RC1)

**General comments**

The manuscript provides a comprehensive analysis of air pollutant emissions resulting from the open burning of various types of household or municipal solid waste. The authors have conducted laboratory combustion experiments and presented detailed chemical speciation data for $PM_{2.5}$, acidic and alkali gases, as well as non-polar organic compounds such as PAHs, nitro-PAHs, alkanes, alkenes, and phthalates. The manuscript's strengths lie in the extensive data collection and thorough analysis. It offers localized emission factors for a variety of waste materials specific to South Africa, which is a notable advancement for air quality management and research in the region. The manuscript addresses a critical knowledge gap in emission inventories, particularly in the context of developing countries where open burning of waste is frequent. If the authors address the following comments sufficiently, the manuscript has the potential to make a significant contribution to the field and would be a valuable addition to the ACP journal.

**Specific comments**

The authors are encouraged to provide additional details and clarifications on the methodology, particularly regarding the selection of waste materials for combustion and the conditions under which the experiments were conducted; see my comments below. This will enhance the reproducibility of the study and improve readability without the need to extensively refer the readers to the companion paper by Wang et al.

o   Regarding the collection of food discards and vegetation samples in Nevada to avoid potential degradation during transportation (Line 84), I am curious as to whether these samples accurately represent the conditions in South Africa.

o   Could you please provide more details on the method used to determine moisture content (Line 89)?

o   In the section describing the preparation of waste materials for testing, it is not entirely clear why each specific step (drying, rehydrating, and re-equilibrating) is necessary. Could you please elaborate on the purpose of these procedures (Lines 89-91)?

o   I noticed that there is a considerable range in the fuel mass used in the experiments, varying from 0.5 to 20 g (Line 97). I am concerned that this variability could potentially influence the emission factors, particularly if the surface area or geometry of the samples plays a significant role in combustion. Could you please address this issue?

o   It would be helpful if you could provide the burn duration in minutes (Line 97), as this unit of measurement may be more intuitive for readers.

o   I am interested in understanding whether the temperature of 450 °C used in the experiments (Line 97) is indicative of smoldering combustion; does your setup allow control over combustion efficiency, which seems to depend on fuel type? Was the modified combustion efficiency calculated based on measurements from gas analyzers, and were these measurements consistent across different materials, especially those that were subjected to repeated testing (Lines 103-104)? How do emission factors and PM abundances depend on ignition temperature?

o When you mention that some materials exhibited both flaming and smoldering phases (Line 99), does this mean that the average fire condition was a mix of both? Were these phases visually confirmed or inferred from MCE calculations?

o Lines 132-134: "*The multiwavelength measurement allowed separation of light absorption by black carbon (BC) from brown carbon (BrC), which has unique wavelength dependence based on fuel and combustion conditions (Chow et al., 2015b; 2018; 2021)*". Does this statement contribute to the manuscript, or can it be omitted?

o Line 153: please provide the abundances of OC and EC from the referenced studies, and then draw comparisons with the data from your own research.

o Line 156: it would enhance the clarity of the figure caption if a direct reference were made to Fig. S1b and the related content in the SI, specifically Text S1, lines S 23-34.

o Line 156: please cite a study to support the choice of "*(organic matter= OC × 1.4)*" and justify its use (is it because you study primary organic emissions?).

o Line 158: refer the readers to the SI text (Text S1) to support the equation: "*minerals = 2.2×Al + 2.49×Si + 1.63×Ca + 2.42×Fe + 1.94×Ti)*".

**Technical correction**

o Line 255: please correct the unit (g kg$^{-1}$).

---

## Referee Comment (RC2)

Re: Chemically Speciated Air Pollutant Emissions from Open Burning of Household Solid Waste from South Africa
Wang et.al.

This paper addresses a major research gap in air quality research in Africa. Use of emission factors derived from Emissions from North America for air quality modeling in Africa has been a serious handicap. The paper provides a very important and critical information for the region and can encourage similar work in other African Countries.

Main concerns that need to be addressed
1. "Open burning has low burning efficiency" needs to be backed by measurements of Modified Combustion Efficiency (MCE) to determine burning conditions.
2. Line 30 Important recent relevant studies are missing and my need to be included (more are added below related to calculation of emission factors and Africa relevant work

Gordon et.al "The Effects of Trash, Residential Biofuel, and Open Biomass Burning Emissions on Local and Transported PM$_{2.5}$ and Its Attributed Mortality in Africa"
https://doi.org/10.1029/2022GH000673

Pokhrel et al. Determination of Emission Factors of Pollutants From Biomass Burning of African Fuels in Laboratory Measurements https://doi.org/10.1029/2021JD034731

Hodshire et. al. "Aging Effects on Biomass Burning Aerosol Mass and Composition: A Critical Review of Field and Laboratory Studies" https://doi.org/10.1021/acs.est.9b02588

3. Line 83: How valid is using food discards from Nevada to be used to represent food discards in Africa. The food discards in Africa are probable fresh from the farm or bakery unlike the processed food with preservative chemicals in the US. How would the preservatives contaminate the samples?
4. Line 97: More details on the burning condition is needed. If a tube furnace is used at 450 it often corresponds to smoldering combustion based on the MCE. Pokhrel et al has shown MCE dependence of emission factors.
5. Some details need to be provided on how trash burning experiments are done. The trash in trash damps in Africa are a mixture of food discards, plastics, paper products and vegetation. How is this exactly done? Furthermore, there is evidence of fuel type dependent emission factors for biomass fuel are reported. When the authors indicate vegetation, it is quite broad, and the type of vegetation needs to be described. The results from the combined waste do not quite match with the results of individual types of trash. If the combination of fuels or trash contains everything, then all the EF's pollutants should show in proportional amounts. How do the authors explain this?
6. The major concern is a missing information on how Emission factors are calculated for each species. Table 1 is an important table, and I am sure all the authors these results are compared to have provided the methods and assumptions used in calculating emission factors. Examples are Pokhrel et al,

Yokelson, R. J., J. G. Goode, D. E. Ward, R. A. Susott, R. E. Babbitt, D. D. Wade, I. Bertschi, D. W. T. Griffith, and W. M. Hao (1999), Emissions of formaldehyde, acetic acid, methanol, and other trace gases from biomass fires in North Carolina measured by airborne Fourier transform infrared spectroscopy, Journal of Geophysical Research-Atmospheres, 104(D23), 30109-30125, doi:10.1029/1999jd900817.

Andreae, M. O., and P. Merlet (2001), Emission of trace gases and aerosols from biomass burning, Global Biogeochemical Cycles, 15(4), 955-966, doi:10.1029/2000gb001382.

Selimovic, V., Yokelson, R. J., Warneke, C., Roberts, J. M., de Gouw, J., Reardon, J., & Griffith, D. W. T. (2018). Aerosol optical properties and trace gas emissions by PAX and OP-FTIR for laboratory-simulated western US wildfires during FIREX. *Atmospheric Chemistry and Physics*, 18(4), 2929–2948. https://doi.org/10.5194/acp-18-2929-2018

Weyant, C. L.; Chen, P.; Vaidya, A.; Li, C.; Zhang, Q.; Thompson, R.; Ellis, J.; Chen, Y.; Kang, S.; Shrestha, G. R.; et al. Emission measurements from traditional biomass cookstoves in south Asia and Tibet. *Environ. Sci. Technol.,* 2019, *53* (6), 3306-3314. DOI: 10.1021/acs.est.8b05199.

Stockwell, C. E., Jayarathne, T., Cochrane, M. A., Ryan, K. C., Putra, E. I., Saharjo, B. H., et al. (2016). Field measurements of trace gases and aerosols emitted by peat fires in Central Kalimantan, Indonesia, during the 2015 El Niño. *Atmospheric Chemistry and Physics*, 16(18), 11711–11732. https://doi.org/10.5194/acp-16-11711-2016

Vakkari, V.; Beukes, J. P.; Dal Maso, M.; Aurela, M.; Josipovic, M.; van Zyl, P. G. Major secondary aerosol formation in southern African open biomass burning plumes. *Nat. Geosci.,* 2018, *11* (8), 580-583. DOI: 10.1038/s41561-018-0170-0.

Minor comments:
Line 10. Is household trash burning a large source of pollutant worldwide or Global South. Developing countries in current literature is now referred to as Global South
Line 12: what does activity data mean?
Line 13: Scarcer? Is it grammatically, correct?
Line 21: Plastic bottles, plastic bags, rubber and .. (remove "and between plastic bottles and bags)
Line 30: Global south instead of developing countries
Line 35-36: Instead of communities with low socioeconomic status better use Low-income communities
Line 42" emission factor and activity data? What is activity data?
Line 58: …highlighted a large variation instead of the
Line 60: Detailed PM chemical composition data are.. (data is missing)
Line 65: PM light scattering, and absorsion properties depend on its chemical composition and associate hygroscopicity and optical properties. change to "PM optical properties depend on chemical composition and hygroscopicity" absorption and scattering are the optical properties
Line 77: Ef's for acidic… remove including elements
Line 244: Higher combustion temperature doesn't indicate burning condition. Need MCE

Line 385: dearth of measurements "of is missing"

---

## Author Comment (AC1)

**Responses to Reviewer 1's Comments**

We thank Reviewer 1's positive comments and constructive suggestions for improving the manuscript. Please see our response in blue font below.

**General Comments**

1. *The authors are encouraged to provide additional details and clarifications on the methodology, particularly regarding the selection of waste materials for combustion and the conditions under which the experiments were conducted.*
   **Response**: Additional information on the methodology, especially on waste materials and test conditions, is added and described in the responses to the reviewer's comments below.

2. *Regarding the collection of food discards and vegetation samples in Nevada to avoid potential degradation during transportation (Line 84), I am curious as to whether these samples accurately represent the conditions in South Africa.*
   **Response**: We tried our best to match the food discards and vegetation samples collected in Nevada with the materials in South Africa. The following text is added in Section 2.1:

   "Due to customs restriction and potential deterioration during shipping, the compositions of food discards and vegetation collected by the WCI were characterized and similar mixtures were collected in Nevada for testing. Food discards included bread, potato and banana peels, lettuce, cucumbers, and tomatoes (Cronjé et al., 2018) and vegetation included basin wild rye, Sandberg bluegrass, crested wheat grass, red willows, and creeping wild rye, representing African bunch grasses, African sumac, and crab grass."

3. *Could you please provide more details on the method used to determine moisture content (Line 89)?*
   **Response**: The following text about the moisture content determination method is added:

   "The moisture contents for paper, leather/rubber, textile, and plastics were determined by a laboratory in South Africa by measuring the mass loss gravimetrically after heating a small fraction of samples at 103 °C for 30 minutes. The moisture contents for food and vegetation were determined at DRI by baking the samples at 90 °C for 24 hours."

4. *In the section describing the preparation of waste materials for testing, it is not entirely clear why each specific step (drying, rehydrating, and re-equilibrating) is necessary. Could you please elaborate on the purpose of these procedures (Lines 89-91)?*
   **Response**: The material moisture content will likely change during transport and storage. Therefore, we measured the natural moisture content immediately after material collection from the field. To return the materials back to this natural moisture condition before testing, we needed to first dry the materials, calculate the water that needs to be added to the dry material to achieve the needed moisture content, and equilibrate for ≥24 hours so that the moisture could assimilate into the materials. A similar procedure has been used in past

studies on the effects of moisture content in source emissions (e.g., Chen et al., 2010; Smith et al., 2013; Watson et al., 2019). The text was modified as below:

"Because material moisture content will likely change during transport and storage, to represent field conditions, the waste materials (except food discards) were oven dried at 90 °C for 24 hours, rehydrated to their natural moisture levels with distilled deionized water (DDW), and re-equilibrated for at least 24 hours before testing."

5. *I noticed that there is a considerable range in the fuel mass used in the experiments, varying from 0.5 to 20 g (Line 97). I am concerned that this variability could potentially influence the emission factors, particularly if the surface area or geometry of the samples plays a significant role in combustion. Could you please address this issue?*
**Response**: We initially planned to use 10 grams of materials for each test. During trial tests, we found that some materials generated very high particulate emissions (e.g., plastic bottles) that clogged the sampling system while some materials generated low gas emissions (e.g., food discards). The weights of materials burned were adjusted to generate emissions that are within the instrument ranges.

We agree that surface area and geometry of the sample affect emissions. The companion paper (Wang et al., 2023) recognizes this limitation and cautions that the lab results might need be adjusted for real-world emissions:

"Real-world open burning emissions vary with waste material composition, pile size, packing structure, moisture content, ambient temperature, and wind speed. Such variations are reflected in the wide range of EFs reported in the literature. Although this and past studies agree within reported extremes, laboratory tests are an approximation of real-world variations. The EFs derived from laboratory experiments represent the values obtained under the specific conditions in laboratory tests; adjustment might be needed when real-world burning conditions are very different from laboratory test conditions."

We added the following bullet point to the Conclusion and Discussion section:

"(5) Results were obtained from laboratory tests simulating real-world conditions. The EFs might need to be adjusted when real-world burning conditions (e.g., moisture content, temperature, and wind) differ significantly from the test conditions used in this study."

6. *It would be helpful if you could provide the burn duration in minutes (Line 97), as this unit of measurement may be more intuitive for readers.*
**Response**: Changed as suggested:

"Each burn typically took 30 min, varying from 15 to 65 min."

7. *I am interested in understanding whether the temperature of 450 °C used in the experiments (Line 97) is indicative of smoldering combustion; does your setup allow control over combustion efficiency, which seems to depend on fuel type? Was the modified combustion efficiency calculated based on measurements from gas analyzers, and were these*

*measurements consistent across different materials, especially those that were subjected to repeated testing (Lines 103-104)? How do emission factors and PM abundances depend on ignition temperature?*

**Response**: The 450 °C heating temperature was selected to simulate the heating of the sample by surrounding materials in an open burning pile. This temperate is somewhat subjective, but it was based on an earlier study showing a transition from low to high mass loss between 450 and 500 °C for a range of materials including textiles and Teflon (Mulholland et al., 2015). Other than maintaining the crucible at 450 °C, the combustion efficiency was not controlled. The modified combustion efficiencies were calculated based on real-time carbon dioxide ($CO_2$) and carbon monoxide (CO) measurements for all tests and the data were reported in Table 2 of the companion paper (Wang et al., 2023). The relative standard deviations of repeated tests were <10% for all materials, indicating consistent test conditions. For flammable waste materials (i.e., paper, textile, plastic bags, dry and natural moist vegetations, and combined wastes), the combustion was ignited by an electric heat gun or a butane lighter. We did not vary the ignition temperature or the crucible heating temperature, so the dependences of PM abundance and emission factor on temperature is not known.

8. *When you mention that some materials exhibited both flaming and smoldering phases (Line 99), does this mean that the average fire condition was a mix of both? Were these phases visually confirmed or inferred from MCE calculations?*

   **Response**: For flaming materials (i.e., paper, textile, soft plastic bags, vegetations with dry and natural moisture contents, and combined waste), they all had both flaming and smoldering phases. The splits between the flaming and smoldering phases were determined by visual observation from the burning videos as well as from the MCE time series. For gases and particles that are measured by real-time instruments, emission factors for flaming and smoldering phases were reported separately in Table 2 of the companion paper (Wang et al., 2023). As the chemical data were collected from integrated samples, only emission factors of the entire burns were reported.

9. *Lines 132-134: "The multiwavelength measurement allowed separation of light absorption by black carbon (BC) from brown carbon (BrC), which has unique wavelength dependence based on fuel and combustion conditions (Chow et al., 2015b; 2018; 2021)". Does this statement contribute to the manuscript, or can it be omitted?*

   **Response**: Large variations of filter colors were observed for filters collected from burning of each material, indicating differences in chemical composition and optical properties. This was presented in Figure 6 of the companion paper. The black and brown carbon will be presented in a future paper on particle optical properties from waste burning. Therefore, we prefer to leave the method description here.

10. *Line 153: please provide the abundances of OC and EC from the referenced studies, and then draw comparisons with the data from your own research.*

    **Response**: The OC and EC abundances from referenced studies are added:

    "High OC and EC abundances were also found for $PM_{2.5}$ from waste burning in other studies. For example, Jayarathne et al. (2018) found average OC and EC abundances of 77% (ranging

59–114%) and 2.6% (ranging 0–12%), respectively, for mixed waste in Nepal. Wu et al. (2021) found carbonaceous components were 80.5–91.4% of $PM_{2.5}$ for flaming burning of various plastics in China, with OC and EC ranging 45–63% and 7–53%, respectively, which are similar to the flaming emissions in this study."

11. *Line 156: it would enhance the clarity of the figure caption if a direct reference were made to Fig. S1b and the related content in the SI, specifically Text S1, lines S 23-34.*
    **Response**: A direct reference to S1 is added in the Figure 2 caption:
    "See detailed description of the major composition categories in Supplemental Materials S1."

12. *Line 156: please cite a study to support the choice of "(organic matter= OC × 1.4)" and justify its use (is it because you study primary organic emissions?).*
    **Response**: The organic mass (OM) to organic carbon (OC) ratio varies with the composition of OM, ranging from 1.2 for fresh vehicle engine emissions (Kleeman et al., 2000) and fresh urban aerosols (Chow et al., 2002) to 2.6 for aged aerosols (Turpin and Lim, 2001). A value of 1.4 has been most commonly used for urban aerosols, and a value of 1.8 is used for more aged non-urban aerosols (Chow et al., 2015). Reid et al. (2005) found the ratio to be ~1.5 for fresh biomass burning smoke, cautioning that this value is highly uncertain. We calculated the OM/OC ratio assuming 100% mass closure for each test condition, and took the average value of 1.4 as the final multiplier. This factor accounts for unmeasured organic elements (e.g., hydrogen, oxygen, and nitrogen) and resulted in reasonable mass closure as shown in Figures 2 and S1b. The following explanation is added to Supplemental S1:

    "The multiplier ($f_{OM/OC}$) for converting OC to OM varies with the composition of OM, ranging from 1.2 for fresh vehicle engine emissions (Kleeman et al., 2000) and fresh urban aerosols (Chow et al., 2002) to 2.6 for aged aerosols (Turpin and Lim, 2001). A value of 1.4 has been most commonly used for urban aerosols, and a value of 1.8 is used for more aged non-urban aerosols (Chow et al., 2015). Reid et al. (2005) found the $f_{OM/OC}$ to be ~1.5 for fresh biomass burning smoke. Assuming that all species are measured and analytical uncertainties are negligible, $f_{OM/OC}$ values for different materials are estimated from mass closure as (Pani et al., 2019):

$$f_{OM/OC} = \frac{PM_{2.5} - EC - Ions - Minerals - Others}{OC} \tag{S1}$$

Table S1 shows that $f_{OM/OC}$ varies from 1.22 for dry vegetation to 1.87 for food discards, with smoldering materials (except rubber) having higher values than flaming fuels, indicating more oxygens in organic aerosols from smoldering combustions. The overall average $f_{OM/OC}$ value for all test conditions is 1.4, which is used to convert OC to OM in mass reconstruction.

Table S1: Measured organic matter (OM) to organic carbon (OC) ratio $f_{OM/OC}$.

| Material | OM/OC |
|---|---|
| Paper | 1.66 ± 0.16 |
| Rubber | 1.27 ± 0.07 |
| Textile | 1.36 ± 0.28 |
| Plastic (Bottles) | 1.42 ± 0.02 |
| Plastic (Bags) | 1.66 ± 0.60 |
| Vegetation (0%) | 1.22 ± 0.11 |
| Vegetation (20%) | 1.38 ± 0.15 |
| Vegetation (50%) | 1.63 ± 0.12 |
| Food Discards | 1.87 ± 0.09 |
| Combined | 1.40 ± 0.21 |

"

13. *Line 158: refer the readers to the SI text (Text S1) to support the equation: "minerals = 2.2×Al + 2.49×Si + 1.63×Ca + 2.42×Fe + 1.94×Ti)".*
    **Response**: Revised as suggested.

14. *Line 255: please correct the unit (g kg$^{-1}$).*
    **Response**: Revised as suggested.

**References**

Chen, L.-W.A., Verburg, P., Shackelford, A., Zhu, D., Susfalk, R., Chow, J.C., Watson, J.G. (2010). "Moisture effects on carbon and nitrogen emission from burning of wildland biomass." *Atmospheric Chemistry and Physics* 10:6617-6625. doi:10.5194/acp-10-6617-2010

Chow, J.C., Watson, J.G., Edgerton, S.A., Vega, E. (2002). "Chemical composition of PM2.5 and PM10 in Mexico City during winter 1997." *Science of the Total Environment* 287 (3):177-201. https://doi.org/10.1016/S0048-9697(01)00982-2

Chow, J.C., Lowenthal, D.H., Chen, L.-W.A., Wang, X.L., Watson, J.G. (2015). "Mass reconstruction methods for PM$_{2.5}$: a review." *Air Quality, Atmosphere & Health* 8 (3):243-263. https://doi.org/10.1007/s11869-015-0338-3

Cronjé, N., Van Der Merwe, I., Müller, I.-M. (2018). "Household food waste: A case study in Kimberley, South Africa." *Journal of Consumer Sciences* 46:1-9.

Jayarathne, T., Stockwell, C.E., Bhave, P.V., Praveen, P.S., Rathnayake, C.M., Islam, M.R., Panday, A.K., Adhikari, S., Maharjan, R., Goetz, J.D., DeCarlo, P.F., Saikawa, E., Yokelson, R.J., Stone, E.A. (2018). "Nepal Ambient Monitoring and Source Testing Experiment (NAMaSTE): emissions of particulate matter from wood- and dung-fueled cooking fires, garbage and crop residue burning, brick kilns, and other sources." *Atmospheric Chemistry and Physics* 18 (3):2259-2286. https://doi.org/10.5194/acp-18-2259-2018

Kleeman, M.J., Schauer, J.J., Cass, G.R. (2000). "Size and composition distribution of fine particulate matter emitted from motor vehicles." *Environmental Science & Technology* 34 (7):1132-1142. 10.1021/es981276y

Mulholland, G.W., Meyer, M., Urban, D.L., Ruff, G.A., Yuan, Z.-g., Bryg, V., Cleary, T., Yang, J. (2015). "Pyrolysis Smoke Generated Under Low-Gravity Conditions." *Aerosol Science and Technology* 49 (5):310-321. 10.1080/02786826.2015.1025125

Pani, S.K., Chantara, S., Khamkaew, C., Lee, C.-T., Lin, N.-H. (2019). "Biomass burning in the northern peninsular Southeast Asia: Aerosol chemical profile and potential exposure." *Atmospheric Research* 224:180-195. https://doi.org/10.1016/j.atmosres.2019.03.031

Reid, J.S., Koppmann, R., Eck, T.F., Eleuterio, D.P. (2005). "A review of biomass burning emissions part II: intensive physical properties of biomass burning particles." *Atmospheric Chemistry and Physics* 5 (3):799-825. 10.5194/acp-5-799-2005

Smith, A.M.S., Tinkham, W.T., Roy, D.P., Boschetti, L., Kremens, R.L., Kumar, S.S., Sparks, A.M., Falkowski, M.J. (2013). "Quantification of fuel moisture effects on biomass consumed derived from fire radiative energy retrievals." *Geophysical Research Letters* 40 (23):6298-6302. https://doi.org/10.1002/2013GL058232

Turpin, B.J., Lim, H.-J. (2001). "Species Contributions to $PM_{2.5}$ Mass Concentrations: Revisiting Common Assumptions for Estimating Organic Mass." *Aerosol Science and Technology* 35 (1):602 - 610.

Wang, X.L., Firouzkouhi, H., Chow, J.C., Watson, J.G., Ho, S.S.H., Carter, W., De Vos, A.S.M. (2023). "Characterization of gas and particle emissions from open burning of household solid waste from South Africa." *Atmospheric Chemistry and Physics* 23:8921–8937. https://doi.org/10.5194/acp-23-8921-2023

Watson, J.G., Cao, J.J., Chen, L.-W.A., Wang, Q.Y., Tian, J., Wang, X.L., Gronstal, S.B., Ho, S.S.H., Watts, A.C., Chow, J.C. (2019). "Gaseous, $PM_{2.5}$ mass, and speciated emission factors from laboratory chamber peat combustion." *Atmospheric Chemistry and Physics* 19 (22):14173-14193. https://doi.org/10.5194/acp-19-14173-2019

Wu, D., Li, Q., Shang, X., Liang, Y., Ding, X., Sun, H., Li, S., Wang, S., Chen, Y., Chen, J. (2021). "Commodity plastic burning as a source of inhaled toxic aerosols." *Journal of Hazardous materials* 416:125820. https://doi.org/10.1016/j.jhazmat.2021.125820

---

## Author Comment (AC2)

**Responses to Reviewer 2's Comments**

We appreciate Reviewer 2's recognition of the importance of this work. The comments are very helpful. Our detailed response to each comment is provided below in blue font. The red font shows modifications to the original text.

**Main Comments:**

1. *"Open burning has low burning efficiency" needs to be backed by measurements of Modified Combustion Efficiency (MCE) to determine burning conditions.*
   **Response**: The Intergovernmental Panel on Climate Change (IPCC, 2006) uses an oxidation factor (the fraction of material carbon that is fully oxidized to $CO_2$) as an indicator of combustion efficiency in the estimation of solid waste burning emissions. The oxidation factor is near 100% for MSW incineration and 58% for open burning. We added this information and also added a reference (Velis and Cook, 2021) for the low combustion efficiencies for open burning:

   "While MSW incineration oxidizes nearly all fuel carbon to carbon dioxide ($CO_2$), open burning only fully oxidizes about 58% of the materials (IPCC, 2006). Open burning has lower combustion efficiencies due to inefficient mixing of fuels and oxygen and low burning temperatures, resulting in emissions of a wide range of air pollutants (Velis and Cook, 2021)."

2. *Line 30 Important recent relevant studies are missing and may need to be included (more are added below related to calculation of emission factors and Africa relevant work*
   - *Gordon et.al "The Effects of Trash, Residential Biofuel, and Open Biomass Burning Emissions on Local and Transported PM2.5 and Its Attributed Mortality in Africa" https://doi.org/10.1029/2022GH000673*
   - *Pokhrel et al. Determination of Emission Factors of Pollutants From Biomass Burning of African Fuels in Laboratory Measurements https://doi.org/10.1029/2021JD034731*
   - *Hodshire et. al. "Aging Effects on Biomass Burning Aerosol Mass and Composition: A Critical Review of Field and Laboratory Studies" https://doi.org/10.1021/acs.est.9b02588*

   **Response**: Thanks for bringing these references to our attention. We have included them in various locations of the manuscript.

   "It is estimated that exposure to $PM_{2.5}$ from open burning of solid waste causes at least 270,000 premature deaths globally (Williams et al., 2019) and 10,000–20,000 premature deaths in Africa (Gordon et al., 2023; Kodros et al., 2016) each year."

   "Pokhrel et al. (2021) reported PM (≤720 nm) EFs for seven types of African woody biomasses, averaging 19.2 (ranging 13.2–25.1) g $kg^{-1}$ for burns with MCE <0.85, which is lower than that for the 50% moisture sample in this study; on the other hand, PM EFs were 5.0 (ranging 0.82–22.2) g $kg^{-1}$ for burns with MCE≥0.85, which are similar to those for the dry and 20% moisture samples."

"Low EFs for particulate Cl⁻, NO₃⁻, SO₄²⁻, and NH₄⁺ were also reported by Pokhrel et al. (2021) for African biomass burning emissions."

"Results were obtained from laboratory tests simulating real-world conditions. However, the differences in fuel mixtures, packing structure, moisture content, burn conditions, dilution, and aging between laboratory and field conditions will cause differences in chemical compositions and EFs (Hodshire et al., 2019). The EFs might need to be adjusted when real-world burning conditions differ significantly from the test conditions used in this study."

3. *How valid is using food discards from Nevada to be used to represent food discards in Africa. The food discards in Africa are probable fresh from the farm or bakery unlike the processed food with preservative chemicals in the US. How would the preservatives contaminate the samples?*

   **Response**: We matched the food discards collected in Nevada with materials burned in South Africa. Our South Africa collaborator categorized the typical composition of food wastes in their solid waste collection, and we collected similar materials in Nevada. The picture below shows the food discards used in our test, which was included in Figure S1 of the companion paper (Wang et al., 2023). All the ingredients were fresh vegetables and bakery products. We believe there is little contamination by preservative chemicals for the materials tested.

[Figure]

Figure S1.h of (Wang et al., 2023): Photograph of food waste materials used in this study.

The following text is added in Section 2.1:

"Due to customs restriction and potential deterioration during shipping, the compositions of food discards and vegetation collected by the WCI were characterized and similar mixtures were collected in Nevada for testing. Food discards included bread, potato and banana peels, lettuce, cucumbers, and tomatoes (Cronjé et al., 2018) and vegetation included basin wild rye, Sandberg bluegrass, crested wheat grass, red willows, and creeping wild rye, representing African bunch grasses, African sumac, and crab grass."

4. *Line 97: More details on the burning condition is needed. If a tube furnace is used at 450 it often corresponds to smoldering combustion based on the MCE. Pokhrel et al has shown MCE dependence of emission factors.*

   **Response**: The combustion experiment is described in detail in Section 2.2 Combustion Experiments of the companion paper (Wang et al., 2023), so only a brief description is provided in this paper. Figures 3, 4, S4, S8, S11, S14, S17, S20-S22, S26, and S29 of the

companion show the time series of MCEs determined from testing of each waste material, including the periods designated as flaming or smoldering combustion. Table 2 of the companion paper lists the mean MCEs for the flaming and smoldering phases as well as the entire burns, and Figure S3 shows the EFs for $CO_2$, CO, $NO_x$ and $PM_{2.5}$ as a function of MCEs. Due to the very different waste material properties, a consistent relation between EFs and MCEs was not observed.

5. *Some details need to be provided on how trash burning experiments are done. The trash in trash damps in Africa are a mixture of food discards, plastics, paper products and vegetation. How is this exactly done? Furthermore, there is evidence of fuel type dependent emission factors for biomass fuel are reported. When the authors indicate vegetation, it is quite broad, and the type of vegetation needs to be described. The results from the combined waste do not quite match with the results of individual types of trash. If the combination of fuels or trash contains everything, then all the EF's pollutants should show in proportional amounts. How do the authors explain this?*
   **Response**: As described in Section 2.1, we tested emissions from nine individual waste categories as well as the combined materials by mixing all categories based on their mass fractions representative of MSW in South Africa township. The combustion experiment, including pictures of the waste materials before and after burning, was described in detail in Section 2.2 Combustion Experiments of the companion paper (Wang et al., 2023).

   As described in the response to Comment 3, the vegetation used in this study included: basin wild rye, Sandberg bluegrass, and crested wheat grass representing African bunch grasses; red willows representing African sumac; and creeping wild rye representing crab grass. We acknowledge that vegetation includes many more varieties, and the derived emission factors apply to the materials reported in this manuscript. The companion paper shows that our 0% and 20% moisture vegetation EFs for $CO_2$, CO, and $SO_2$ were in good agreement with those derived for Savanna vegetation (Akagi et al., 2011), while the $PM_{2.5}$ EFs for 50% moisture vegetation burning were about one order of magnitude higher than literature values.

   We recognize that the mass-weighted sum of EFs from individual waste material does not equate to the combined materials EFs. The companion paper cautioned the readers for using separate or combined emission factors in Section 3.5 as follows (Wang et al., 2023):

   "However, it should be cautioned that the burning behaviors differ between separated and combined waste materials, causing emissions to change. Table S5 compares the measured EFs for the combined materials and the values calculated from $EF_{p,i}$. The calculated EFs agree with the measured values within 10% for $CO_2$ and $NO_x$; however, the calculated EFs for CO and PM are over 50% and 600% higher, respectively. It is possible that more efficient combustion in the combined materials lowered CO and PM emissions as compared to less efficient individual burns, particularly for materials that only smoldered and had high EFs for CO and PM. Additionally, laboratory measured $EF_{p,i}$ or $EF_p$ might differ from field values given the complex waste mixtures and burning conditions. Adjustments to laboratory $EF_{p,i}$ might be needed when estimating real-world $EF_p$. Future studies comparing in situ measurement from a variety of representative real-world burns with laboratory data would assist in establishing adjustment factors."

6.  *The major concern is missing information on how Emission factors are calculated for each species. Table 1 is an important table, and I am sure all the authors these results are compared to have provided the methods and assumptions used in calculating emission factors Pokhrel et al. and other references.*
    *Examples are*

    *Yokelson, R. J., J. G. Goode, D. E. Ward, R. A. Susott, R. E. Babbitt, D. D. Wade, I. Bertschi, D. W. T. Griffith, and W. M. Hao (1999), Emissions of formaldehyde, acetic acid, methanol, and other trace gases from biomass fires in North Carolina measured by airborne Fourier transform infrared spectroscopy, Journal of Geophysical Research-Atmospheres, 104(D23), 30109-30125, doi:10.1029/1999jd90081*

    *Andreae, M. O., and P. Merlet (2001), Emission of trace gases and aerosols from biomass burning, Global Biogeochemical Cycles, 15(4), 955-966, doi:10.1029/2000gb001382.*

    *Selimovic, V., Yokelson, R. J., Warneke, C., Roberts, J. M., de Gouw, J., Reardon, J., & Griffith, D. W. T. (2018). Aerosol optical properties and trace gas emissions by PAX and OP-FTIR for laboratory-simulated western US wildfires during FIREX. Atmospheric Chemistry and Physics, 18(4), 2929–2948. https://doi.org/10.5194/acp-18-2929-201*

    *Weyant, C. L.; Chen, P.; Vaidya, A.; Li, C.; Zhang, Q.; Thompson, R.; Ellis, J.; Chen, Y.; Kang, S.; Shrestha, G. R.; et al. Emission measurements from traditional biomass cookstoves in south Asia and Tibet. Environ. Sci. Technol., 2019, 53 (6), 3306-3314. DOI: 10.1021/acs.est.8b05199.*

    *Stockwell, C. E., Jayarathne, T., Cochrane, M. A., Ryan, K. C., Putra, E. I., Saharjo, B. H., et al. (2016). Field measurements of trace gases and aerosols emitted by peat fires in Central Kalimantan, Indonesia, during the 2015 El Niño. Atmospheric Chemistry and Physics, 16(18), 11711–11732. https://doi.org/10.5194/acp-16-11711-2016*

    *Vakkari, V.; Beukes, J. P.; Dal Maso, M.; Aurela, M.; Josipovic, M.; van Zyl, P. G. Major secondary aerosol formation in southern African open biomass burning plumes. Nat. Geosci., 2018, 11 (8), 580-583. DOI: 10.1038/s41561-018-0170-0.*

    **Response**: Thanks for providing relevant references. Indeed, past studies and approaches to estimate emission factors (EFs) were examined. Our companion paper (Wang et al., 2023) documented the EF calculation in Eq. (2). We made the assumption that fuel carbon emitted as methane and volatile organics is negligible. A unique feature of this study is that we included carbon in the ash and PM in the EF calculation, and evaluated the effects of neglecting these terms. Section 3.4 of Wang et al. (2023) shows that without including ash and/or PM carbon, changes in EFs are <5% for flaming dominated combustions. However, the consequences of not including ash or PM carbon are larger for smoldering fuels. For smoldering plastic bottles, not including carbon in PM resulted in an EF overestimation of 577%; in addition, if ash carbon was not included, the EFs would be overestimated by 623%.

    Data from Andreae (2019) are added in Table 1 because it has EFs for BC and PAHs for biomass and garbage. The other studies are not included because they either do not have BC and PAH EFs or the fuel and burning conditions are different from open burning (e.g., peat or cook stoves).

**Minor comments:**

7. *Line 10. Is household trash burning a large source of pollutant worldwide or Global South. Developing countries in current literature is now referred to as Global South*
   **Response**: As suggested, "developing countries" is replaced as "Global South" in the text. Trash burning is a source of pollution worldwide and in the Global South.

8. *Line 12: what does activity data mean?*
   **Response**: Activity is a term used in emission inventories to reflect emission generation activities. Typical activities include amount of fuel burned, vehicle kilometers traveled, etc. (IPCC, 2006). Emissions are generally estimated as (U.S. EPA, 2017):

   $$E = A \times EF \times (1 - ER / 100)$$

   where E is the total emissions, A is activity indicator, EF is emission factor, and ER is overall emission reduction efficiency in percent. For MSW open burning, the activity data is the amount of waste burned. The text is revised as follows:

   "Despite the large environmental impacts of uncontrolled MSW open burning, its emissions are not included or are poorly represented in local, regional, and global emission inventories due to lack of information on emission factor (EF) and amount of MSW burned (activity) (Cook and Velis, 2021; Ramadan et al., 2022)"

9. *Line 13: Scarcer? Is it grammatically, correct?*
   **Response**: We believe "Detailed particulate matter (PM) chemical speciation data is even scarcer" is grammatically correct. However, the word scarcer is replaced with "less available".

10. *Line 21: Plastic bottles, plastic bags, rubber and .. (remove "and between plastic bottles and bags)*
    **Response**: Revised as suggested.

11. *Line 30: Global south instead of developing countries*
    **Response**: Revised as suggested.

12. *Line 35-36: Instead of communities with low socioeconomic status better use Low-income communities*
    **Response**: Revised as suggested.

13. *Line 42" emission factor and activity data? What is activity data?*
    **Response**: Please see response to Comment 8.

14. *Line 58: ...highlighted a large variation instead of the*
    **Response**: Revised as suggested.

15. *Line 60: Detailed PM chemical composition data are.. (data is missing)*
    **Response**: Revised as suggested.

16. *Line 65: PM light scattering, and absorption properties depend on its chemical composition and associate hygroscopicity and optical properties. change to "PM optical properties depend on chemical composition and hygroscopicity" absorption and scattering are the optical properties*
    **Response**: The overall PM optical properties depend on the optical properties of individual chemical components. We revised the sentence as below to be clearer:

    "PM light scattering and absorption properties depend on the hygroscopicity and optical properties of its chemical components."

17. *Line 77: Ef's for acidic… remove including elements*
    **Response**: The sentence is revised as below:

    "This paper focuses on speciated source profiles and EFs for  elements, acidic and alkali gases and ions, PAHs, nitro-PAHs, n-alkanes, and phthalates."

18. *Line 244: Higher combustion temperature doesn't indicate burning condition. Need MCE*
    **Response**: The MCE information is added:
    "The modified combustion efficiencies (MCEs) for the dry (MCE = 0.88) and 20% moisture (MCE = 0.91) vegetation samples were higher than the 50% moisture vegetation sample (MCE = 0.79) (Wang et al., 2023). One would expect that the dry and 20% moisture vegetation samples would cause higher EFs for HF than particulate $F^-$ due to preferred partition in the gas phase at higher combustion temperatures and MCEs."

19. *Line 385: dearth of measurements "of is missing"*
    **Response**: Revised as suggested.

20. *Line 255: please correct the unit ($g\ kg^{-1}$).*
    **Response**: Revised as suggested.

**References**
Akagi, S.K., Yokelson, R.J., Wiedinmyer, C., Alvarado, M.J., Reid, J.S., Karl, T., Crounse, J.D., Wennberg, P.O. (2011). "Emission factors for open and domestic biomass burning for use in atmospheric models." *Atmospheric Chemistry and Physics* 11 (9):4039-4072. https://doi.org/10.5194/acp-11-4039-2011

Andreae, M.O. (2019). "Emission of trace gases and aerosols from biomass burning – an updated assessment." *Atmospheric Chemistry and Physics* 19 (13):8523-8546. 10.5194/acp-19-8523-2019

Cook, E., Velis, C., (2021). "Global review on safer end of engineered life." Royal Academy of Engineering, London, UK. https://eprints.whiterose.ac.uk/169766/. https://doi.org/10.5518/100/58

Cronjé, N., Van Der Merwe, I., Müller, I.-M. (2018). "Household food waste: A case study in Kimberley, South Africa." *Journal of Consumer Sciences* 46:1-9.

Gordon, J.N.D., Bilsback, K.R., Fiddler, M.N., Pokhrel, R.P., Fischer, E.V., Pierce, J.R., Bililign, S. (2023). "The Effects of Trash, Residential Biofuel, and Open Biomass Burning Emissions on Local and Transported PM2.5 and Its Attributed Mortality in Africa." *GeoHealth* 7 (2):e2022GH000673. https://doi.org/10.1029/2022GH000673

Hodshire, A.L., Akherati, A., Alvarado, M.J., Brown-Steiner, B., Jathar, S.H., Jimenez, J.L., Kreidenweis, S.M., Lonsdale, C.R., Onasch, T.B., Ortega, A.M., Pierce, J.R. (2019). "Aging Effects on Biomass Burning Aerosol Mass and Composition: A Critical Review of Field and Laboratory Studies." *Environmental Science & Technology* 53 (17):10007-10022. 10.1021/acs.est.9b02588

IPCC, (2006). "2006 IPCC guidelines for national greenhouse gas inventories." National Greenhouse Gas Inventories Programme Japan; Intergovernmental Panel on Climate Change (IPCC): Geneva, Switzerland.

Kodros, J.K., Wiedinmyer, C., Ford, B., Cucinotta, R., Gan, R., Magzamen, S., Pierce, J.R. (2016). "Global burden of mortalities due to chronic exposure to ambient PM 2.5 from open combustion of domestic waste." *Environmental Research Letters* 11 (12):124022. https://doi.org/10.1088/1748-9326/11/12/124022

Pokhrel, R.P., Gordon, J., Fiddler, M.N., Bililign, S. (2021). "Determination of Emission Factors of Pollutants From Biomass Burning of African Fuels in Laboratory Measurements." *Journal of Geophysical Research: Atmospheres* 126 (20):e2021JD034731. https://doi.org/10.1029/2021JD034731

Ramadan, B.S., Rachman, I., Ikhlas, N., Kurniawan, S.B., Miftahadi, M.F., Matsumoto, T. (2022). "A comprehensive review of domestic-open waste burning: recent trends, methodology comparison, and factors assessment." *Journal of Material Cycles and Waste Management* 24 (5):1633-1647. https://doi.org/10.1007/s10163-022-01430-9

U.S. EPA, (2017). "Emissions Inventory Guidance for Implementation of Ozone and Particulate Matter National Ambient Air Quality Standards (NAAQS) and Regional Haze Regulations." Air Quality Assessment Division, Office of Air Quality Planning and Standards, U.S. Environmental Protection Agency (U.S. EPA) Research Triangle Park, NC. Accessed on October 29, 2023. https://www.epa.gov/sites/default/files/2017-07/documents/ei_guidance_may_2017_final_rev.pdf.

Velis, C.A., Cook, E. (2021). "Mismanagement of Plastic Waste through Open Burning with Emphasis on the Global South: A Systematic Review of Risks to Occupational and Public Health." *Environmental Science & Technology* 55 (11):7186-7207. https://doi.org/10.1021/acs.est.0c08536

Wang, X.L., Firouzkouhi, H., Chow, J.C., Watson, J.G., Ho, S.S.H., Carter, W., De Vos, A.S.M. (2023). "Characterization of gas and particle emissions from open burning of household solid waste from South Africa." *Atmospheric Chemistry and Physics* 23:8921–8937. https://doi.org/10.5194/acp-23-8921-2023

Williams, M., Gower, R., Green, J., Whitebread, E., Lenkiewicz, Z., Schröder, P., (2019). "No time to waste: Tackling the plastic pollution crisis before it's too late." Tearfund, London, UK. https://opendocs.ids.ac.uk/opendocs/handle/20.500.12413/14490.